# A BAFF/APRIL axis regulates obesogenic diet-driven weight gain

Calvin C. Chan [1,2,3,4,29], Isaac T. W. Harley [1,2,3,4,22,29], Paul T. Pfluger [5,6,7,8], Aurelien Trompette[1,2,23], Traci E. Stankiewicz[1,2], Jessica L. Allen[1,2,4,24], Maria E. Moreno-Fernandez [1,2], Michelle S. M. A. Damen [1,2], Jarren R. Oates [1,2,4], Pablo C. Alarcon [1,2,3,4], Jessica R. Doll [1,2], Matthew J. Flick[1,9,25], Leah M. Flick[1,2,26], Joan Sanchez-Gurmaches [1,10,11], Rajib Mukherjee [1,10], Rebekah Karns[1,12], Michael Helmrath[13,14], Thomas H. Inge [15], Stuart P. Weisberg [16], Sünje J. Pamp[17,18,27], David A. Relman [17,18,19], Randy J. Seeley [20], Matthias H. Tschöp[6,7,8], Christopher L. Karp[1,2,3,4,28] & Senad Divanovic [1,2,3,4,21✉]

The impact of immune mediators on weight homeostasis remains underdefined. Interrogation of resistance to diet-induced obesity in mice lacking a negative regulator of Toll-like receptor signaling serendipitously uncovered a role for B cell activating factor (BAFF). Here we show that overexpression of BAFF in multiple mouse models associates with protection from weight gain, approximating a log-linear dose response relation to BAFF concentrations. Gene expression analysis of BAFF-stimulated subcutaneous white adipocytes unveils upregulation of lipid metabolism pathways, with BAFF inducing white adipose tissue (WAT) lipolysis. Brown adipose tissue (BAT) from BAFF-overexpressing mice exhibits increased *Ucp1* expression and BAFF promotes brown adipocyte respiration and in vivo energy expenditure. A proliferation-inducing ligand (APRIL), a BAFF homolog, similarly modulates WAT and BAT lipid handling. Genetic deletion of both BAFF and APRIL augments diet-induced obesity. Lastly, BAFF/APRIL effects are conserved in human adipocytes and higher BAFF/APRIL levels correlate with greater BMI decrease after bariatric surgery. Together, the BAFF/APRIL axis is a multifaceted immune regulator of weight gain and adipose tissue function.

A list of author affiliations appears at the end of the paper.

Obesity is a major and unabated public health problem[1–3]. Obesity stems from an imbalance between energy intake (e.g., food intake, microbiome, nutrient absorption), energy expenditure (EE) (e.g., basal metabolism, thermogenesis), and energy turnover (lipolysis, lipogenesis)[4,5]. Brown adipose tissue (BAT) and white adipose tissue (WAT) are key contributors to controlling body weight[6,7]. BAT, a highly energy consuming tissue consisting of numerous mitochondria, is capable of removing lipids from circulation to activate adaptive thermogenesis[8]. WAT can assimilate calories through storage of lipids into adipocytes. WAT lipolysis facilitates the breakdown of lipids into functional consumable units including glycerol[9]. Persistent imbalances of caloric intake and EE leads to the pathological expansion of adipose tissue (AT) that contributes to the propagation of obesity-associated low-grade inflammation. This maladaptive obesity-associated inflammation underlies the pathogenesis of many obesity-associated sequelae, including type 2 diabetes mellitus, atherosclerosis, and non-alcoholic fatty liver disease (NAFLD)[10,11]. In light of the clinical and public health challenges, leveraging a better understanding of the pathogenesis of weight gain may lead to new innovative therapeutic approaches[12,13].

Obesity is associated with an increase in circulating Toll-like receptor (TLR) ligands (e.g., lipopolysaccharide [LPS])[14]. However, the contribution of TLRs, including TLR4, to obesity pathogenesis remains controversial with both beneficial[15] and detrimental[16–19] roles reported. TLR4 signaling in myeloid cells is negatively regulated by radioprotective 105 kDa protein (RP105; CD180)[20], a molecule originally thought to be B-cell-specific and facilitate B-cell proliferation[20,21] in response to LPS. Investigation of paradoxical findings that RP105 has dichotomous cell type-specific effects on TLR4 signaling identified that RP105-mediated regulation of B-cell proliferation was not owing to B-cell intrinsic RP105 expression[22]. Rather, chronic upregulation of B-cell activating factor (BAFF) expression was revealed as the mechanism underlying suppressed B-cell proliferation in response to LPS in RP105-deficient mice[22]. Despite its known actions as a negative regulator of TLR-driven inflammation, RP105-deficiency dampens AT inflammation and protects mice from obesity-associated metabolic derangements[23]. However, the mechanisms underlying protection from diet-induced obesity (DIO) in RP105-deficient mice have not been defined.

Both TLR4 and RP105 modulate the production of inflammatory mediators including the tumor necrosis factor (TNF) superfamily members (e.g., TNF, BAFF)[22,24]. TNF can induce white adipocyte lipolysis[25,26] and may modulate brown adipocyte thermogenesis[27–30]. Although recent studies highlight the ability of immune-modulators to regulate AT function[31–34], the link between immune mediators, specifically TNF superfamily members, weight gain, and BAT/WAT function remains poorly understood. BAFF and its close homolog A proliferation-inducing ligand (APRIL) are regulators of B-cell maturation and survival[35]. BAFF and APRIL are produced by diverse myeloid, lymphoid, and non-hematopoietic cell types[36], including adipocytes[37], astrocytes[38], and gut epithelial cells[39]. Human and mouse adipocytes express all known BAFF and APRIL receptors including BAFF receptor (BAFF-R), transmembrane activator and CAML interactor (TACI), and B-cell maturation protein (BCMA).[37] Although BAFF can bind all three receptors, APRIL only binds to TACI and BCMA[35]. Recent reports indicate that BAFF is a regulator of atherosclerosis, glucose dysmetabolism and BAFF-R and TACI regulate obesity-associated metabolic sequelae[40–44]. Importantly, whether and how a BAFF/APRIL axis contributes to the regulation of weight gain in DIO models has not been examined.

Our exploration of mechanisms regulating protection from DIO in RP105-deficient mice led to the serendipitous discovery of a critical role of BAFF and APRIL in the regulation of weight gain. Here, our data indicate that multiple transgenic mouse lines with increase in BAFF expression associates with protection from weight gain. BAFF induces upregulation of lipid metabolism pathways in subcutaneous white adipocytes and is sufficient to augment WAT lipolysis in vivo and in vitro. BAFF is also sufficient to enhance brown adipocyte respiration in vitro and energy expenditure (EE) in vivo. APRIL, a close homolog of BAFF, also modifies both WAT and BAT lipid handling. Lack of both BAFF and APRIL robustly augments DIO in mice. BAFF/APRIL effects are conserved in human adipocytes and higher systemic BAFF/APRIL levels correlate with a greater BMI decline in post-bariatric surgery. Our collective data highlight a novel pathway of immune-mediated regulation of AT/adipocyte lipid handling and may represent novel targets for the treatment of obesity.

## Results

**BAFF modulates diet-induced weight gain.** High-fat diet (HFD)-fed RP105$^{-/-}$ mice exhibited robust protection from DIO, altered subcutaneous (inguinal [iWAT]) and visceral (epididymal [eWAT] and perirenal [pWAT]) adiposity, obesity-associated metabolic sequelae, and enhanced EE as compared with WT counterparts (Fig. 1a–f; Supplementary Fig. 1a–b), in agreement with a previous report[23]. Exploration of direct underlying mechanisms driving unexpected resistance to DIO in RP105$^{-/-}$ mice revealed that the effect was not associated with differential energy intake/metabolism, fat absorption, leptin sensitivity, or intestinal microbiome composition (Supplementary Fig. 1c–f). Employment of bone marrow transfer studies suggested that lack of RP105 expression in the non-hematopoietic compartment was critical for protective effects of RP105 genetic ablation (Supplementary Fig. 2a–b). Generation and utilization of an effective conditional RP105 deletion (RP105$^{flox/flox}$ mice; Supplementary Fig. 2c) in total immune cells (Vav1$^{cre}$), B cells (CD19$^{cre}$), skeletal muscle (MLC$^{cre}$), or central/peripheral nervous system (Nestin$^{cre}$) failed to fully reveal a primary RP105-expressing cellular locus responsible for observed differential weight gain (Supplementary Fig. 2d–g). Collectively, these findings suggested the possibility of an indirect mechanism underlying protection from DIO in RP105$^{-/-}$ mice.

Given the functional relevance of BAFF in B-cell responsiveness in RP105$^{-/-}$ mice[22], we hypothesized that BAFF, which can be induced by multiple proinflammatory means (e.g., TLR signaling[36]), may also affect DIO in RP105$^{-/-}$ mice. Deletion of BAFF in RP105$^{-/-}$ mice (RP105$^{-/-}$/BAFF$^{-/-}$) was sufficient to reverse the protection from HFD-driven weight gain, fasting glucose, glucose tolerance, and fat pad distribution as compared to WT controls (Fig. 1b–e; Supplementary Fig. 3a). Deletion of BAFF in RP105$^{-/-}$ mice was likewise sufficient to dampen EE (Fig. 1f; Supplementary Fig. 3b–c) without alteration to locomotor activity (Supplementary Fig. 3d). These observations indicated that BAFF may regulate weight homeostasis in RP105$^{-/-}$ mice.

To define whether the effect of BAFF on weight gain extends beyond the RP105 system, we utilized multiple model systems that enhance the expression of BAFF. Although RP105$^{-/-}$ mice exhibit marginally increased BAFF expression (~20% increase), B-cell-deficient mice (µMT) manifest a two log-fold secondary increase in BAFF, and BAFF-Tg mice maifest a three log-fold increase in BAFF (Fig. 1g). Enhanced systemic BAFF levels were associated with increased resistance to HFD-driven weight gain in an approximate log-linear dose fashion (Fig. 1g–h). HFD-fed BAFF-Tg (Fig. 1i–m; Supplementary Fig. 4) and µMT (Supplementary Fig. 5) mice were protected from weight-driven glucose dysmetabolism (e.g., fasting glucose and GTT) and NAFLD progression (hepatic triglyceride levels and alanine transaminase [ALT]).

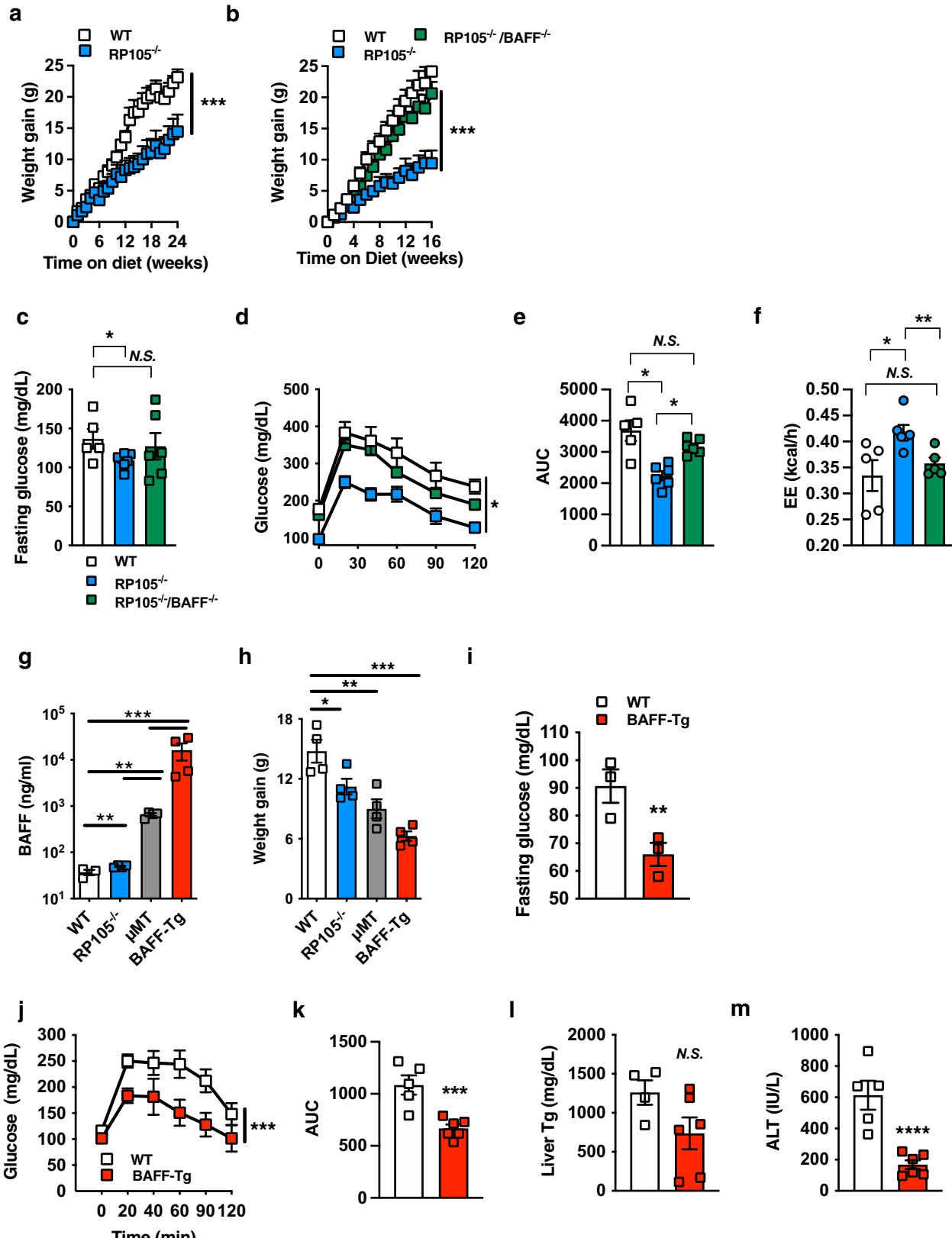

Overall, these findings suggest that increased circulating BAFF levels may broadly regulate DIO-driven weight gain.

**BAFF modifies subcutaneous adipocyte lipolysis.** Obesity pathogenesis/weight gain is regulated, in part, by adipocyte lipid

handling. Inflammatory mediators (e.g., TNF) regulate lipid processing[25,32]. BAFF, a TNF superfamily member, and all three known BAFF receptors (e.g., BAFF-R, TACI, BCMA) are expressed by both mouse and human adipocytes.[37] Thus, we examined the ability of BAFF to modulate adipocyte function.

**Fig. 1 BAFF axis regulates obesity development. a** Weight gain of WT or RP105$^{-/-}$ mice fed a high-fat diet (HFD) for 24 weeks. **b–d** WT, RP105$^{-/-}$ or RP105$^{-/-}$/BAFF$^{-/-}$ mice were fed HFD for 16 weeks. **b** Weight gain. **c** Fasting glucose. **d** Glucose tolerance test (GTT). **e** Area under the curve (AUC). **f** Low-fat chow diet fed WT, RP105$^{-/-}$, or RP105$^{-/-}$/BAFF$^{-/-}$ were monitored in TSE Phenomaster systems for 3 days. Combined energy expenditure. **g–h** WT, RP105$^{-/-}$, μMT or BAFF-Tg mice fed HFD for 15 weeks. **g** Systemic BAFF concentration quantified by ELISA. **h** Weight gain. **i–m** WT or BAFF-Tg mice fed HFD for 20 weeks. **i** Fasting glucose at 20 weeks. **j** GTT at 12 weeks. **k** AUC. **l** Liver triglycerides quantified by colorimetric assay at harvest. **m** Systemic alanine transaminase (ALT) at harvest. **a** Representative of six independent experiments, $n = 4-8$/condition. **b–e** Representative of three independent experiments, $n = 4-7$/condition. **f** A single experiment, $n = 5-6$/condition. **g–h** Representative of three independent experiments, $n = 3-4$/condition. **i–m** Representative of three independent experiments, $n = 3-6$/condition. **a–m** For bar and line graphs, data represents mean±SEM. **a–b**, **d–e**, **j–k** Area under the curve. N.S = not significant, *$p < 0.05$, ***$p < 0.001$. **g–h** One-way ANOVA with Tukey correction. *$p < 0.05$, **$p < 0.01$, ***$p < 0.001$. **c**, **f**, **i**, **l–m**) Unpaired two-tailed Student's $t$ test. *$p < 0.05$, **$p < 0.01$, ***$p < 0.001$, ****$p < 0.0001$. Source data are provided as a Source data file.

Utilization of an unbiased RNA-seq analysis revealed that recombinant BAFF (rBAFF), compared with saline-treated counterparts, robustly upregulated lipid handling pathways, including metabolism of lipids and lipoproteins, regulation of lipid metabolic process, neutral lipid metabolic process, and cellular lipid metabolic process (Fig. 2a) in primary mouse subcutaneous (inguinal) adipocytes. In line with these observations, visceral (epididymal) WAT from RP105$^{-/-}$ and BAFF-Tg mice had significantly increased mRNA expression of adipose triglyceride lipase (*Pnpla2*) and hormone-sensitive lipase (*Lipe*), both critical regulators of lipolysis (Fig. 2b). Enhanced mRNA expression of *Pnpla2* and *Lipe* correlated with increases in systemic BAFF levels (Figs. 1g and 2b). We next tested the direct impact of BAFF on subcutaneous white adipocyte lipolysis. rBAFF was sufficient to augment the presence of free glycerol in adipocyte supernatants, which did not synergize with norepinephrine (Fig. 2c), a major lipolysis-inducing hormone. Further, rBAFF was sufficient to enhance the mRNA expression of both *Pnpla2* and *Lipe* in vitro and in vivo (Fig. 2d, e) and amplify protein expression of phosphorylated hormone-sensitive lipase (HSL; aka Lipe; Fig. 2f). Endotoxin was undetectable in utilized rBAFF (Supplementary Fig. 6a) and heat-inactivated rBAFF did not augment the presence of free glycerol in subcutaneous white adipocytes (Supplementary Fig. 6b). Genetic deletion of BAFF-R, TACI, or BCMA alone abrogated rBAFF-driven subcutaneous adipocyte lipolysis in vitro (Fig. 2g). Similarly, rBAFF did not augment lipolysis in BAFF$^{-/-}$/BCMA$^{-/-}$ or BAFF$^{-/-}$/TACI$^{-/-}$/BCMA$^{-/-}$ subcutaneous adipocytes (Supplementary Fig. 7). rBAFF-driven lipolysis was specific to white, but not brown adipocytes (Supplementary Fig. 8a). Combined, these findings indicate that BAFF is capable of modulating WAT/subcutaneous white adipocyte lipolysis.

**BAFF modifies Ucp1 expression and EE in BAT.** BAT/brown adipocyte thermogenesis is a key regulator of lipid utilization and weight control. We examined whether BAFF could impact BAT thermogenic capacity. Uncoupling protein 1 (Ucp1) promotes adaptive thermogenesis in BAT[45]. BAT from RP105$^{-/-}$ and BAFF-Tg mice, compared with WT controls, exhibited increased *Ucp1* mRNA expression (Fig. 3a). BAT mitochondria from these mice had significantly enhanced respiration as compared with WT controls (Fig. 3b). Congruently, rBAFF was sufficient to amplify *Ucp1* mRNA expression (Fig. 3c), enhance Ucp1-dependent respiration in primary brown adipocytes in vitro (Fig. 3d) and raise the threshold of norepinephrine-driven thermogenic activity (Fig. 3e) in WT mice in vivo. rBAFF enhancement of *Ucp1* mRNA expression was specific to brown adipocytes and not subcutaneous adipocytes (Supplementary Fig. 8b). Subcutaneous WAT from RP105$^{-/-}$, BAFF-Tg, RP105$^{-/-}$/BAFF$^{-/-}$, BAFF$^{-/-}$ mice exhibited similar mRNA expression of Ucp1-dependent thermogenic/beige markers (e.g., *Pgc1a*, *Ucp1*, *Dio2*, *Cidea*, *Elovl6*) as compared with WT counterparts

(Supplementary Fig. 9). We additionally assessed the sufficiency of exogenous BAFF to modify whole-body thermogenic parameters in vivo. In contrast to pre-treatment (Supplementary Fig. 10a), short-term exogenous rBAFF in CD-fed mice transiently amplified EE, specifically in the dark cycles (Supplementary Fig. 10b–c) but did not alter the respiratory exchange ratio or body weight (Supplementary Fig. 10d–e). Congruently, rBAFF administration to HFD-fed mice enhanced total EE (Fig. 3f, g). Collectively, these findings indicate that BAFF, in parallel to its effects on subcutaneous WAT/adipocytes, is capable of modulating BAT/brown adipocyte *Ucp1* expression and in vivo EE.

**APRIL similarly modulates white and brown adipocyte lipid handling.** Examination of HFD-fed BAFF$^{-/-}$ mice revealed no alteration in weight gain (Fig. 4a), WAT or BAT weight (Supplementary Fig. 11a), slightly increased total body weight (Supplementary Fig. 11b), similar fasting glucose (Supplementary Fig. 11c), and protection from hepatocellular injury (Supplementary Fig. 11d)—findings consistent with previous reports[46,47]. Cumulatively, these findings hinted the possibility that other BAFF-like molecules may similarly modulate weight gain to potentially compensate for such effects in the absence of BAFF. BAFF$^{-/-}$ mice exhibited enhanced circulating levels of APRIL (Fig. 4b), a close homolog of BAFF[35]. Congruent with similarities of BAFF and APRIL function on immune cells, an unbiased RNA-seq approach revealed significant gene overlap (52% of BAFF-induced genes) between BAFF or APRIL-treated primary subcutaneous white adipocytes (769 genes; Fig. 4c). Pathway analyses revealed that APRIL stimulation of subcutaneous white adipocytes also upregulated pathways associated with lipolysis including regulation of lipid metabolism, fatty acid metabolism, PPAR signaling, and the glucocorticoid receptor regulatory network (Fig. 4d). rAPRIL utilized did not exhibit endotoxin contamination (Supplementary Fig. 12). Like BAFF, APRIL was sufficient to augment subcutaneous white adipocyte mRNA expression of *Lipe* and *Pnpla2* in vitro and in vivo (Fig. 4e, f) and p-HSL (aka p-Lipe) protein levels in vitro (Fig. 4g). Lack of TACI or BCMA, but not BAFF-R, prohibited APRIL-driven subcutaneous white adipocyte lipolysis in vitro (Fig. 4h)—as would be expected from the pattern of receptor usage by APRIL[35]. Likewise, rAPRIL-mediated lipolysis was blunted in BAFF-R$^{-/-}$/BCMA$^{-/-}$ and BAFF-R$^{-/-}$/TACI$^{-/-}$/BCMA$^{-/-}$ subcutaneous white adipocytes (Supplementary Fig. 13). Examination of APRIL's effects on brown adipocytes demonstrated sufficiency to augment *Ucp1* mRNA expression and enhance mitochondrial respiration (Fig. 4i, j). However, exogenous rAPRIL administration did not amplify EE in lean or obese mice (Supplementary Fig. 14). Together, these findings indicate that APRIL, similar to BAFF, is capable of modulating both subcutaneous white and brown adipocyte lipid handling.

Although close homologs with some overlapping function, deeper examination of our unbiased RNA-seq analyses displayed

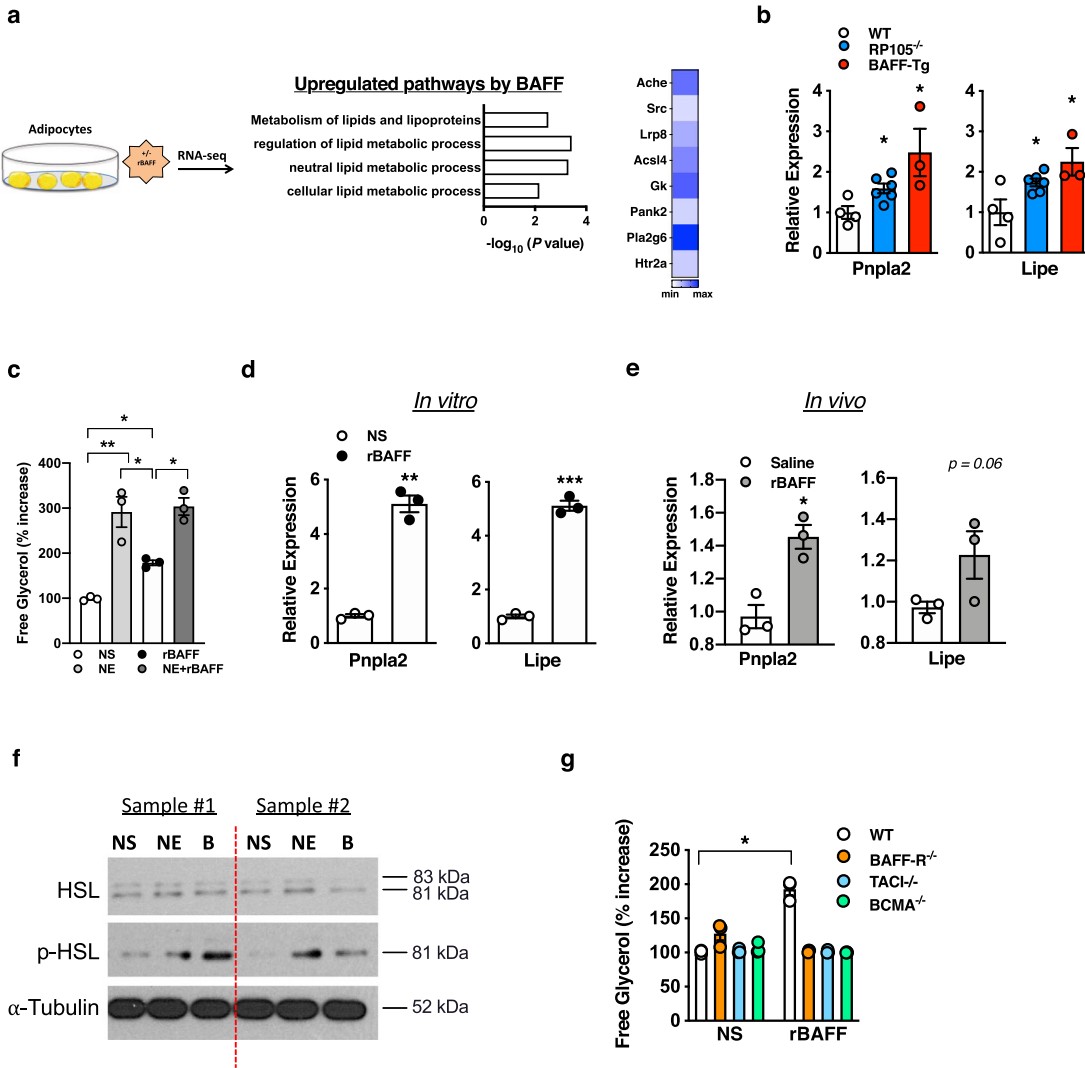

**Fig. 2 BAFF modifies white adipose lipid handling. a** White adipocytes stimulated in the presence or absence of recombinant BAFF (rBAFF; 500 ng/ml) and subjected to RNA-seq analysis. Ontology pathways and heat map of selected associated genes. **b** eWAT mRNA expression quantified by qPCR of indicated lipolysis genes in WT, RP105$^{-/-}$ or BAFF-Tg mice after 20 weeks on a HFD. **c–d** White adipocytes treated with saline (NS), norepinephrine (1 μM), or rBAFF (500 ng/ml) for 24 hours. **c** Free glycerol. % increase relative to WT NS. **d** mRNA expression of indicated lipolysis genes. **e** CD-fed WT mice were treated i.p. with rBAFF (2 μg/mouse) every other day for 1 week. eWAT mRNA expression of indicated lipolysis genes. **f** Primary white adipocytes from WT mice were treated with saline (NS), norepinephrine (NE; 1 μM), or rBAFF (**b** 500 ng/ml) for 12 hours. Protein expression of hormone-sensitive lipase (HSL), phosphorlayted HSL (p-HSL), and α-tubulin loading control by western blot. **g** Free glycerol in supernatants of WT, BAFF-R$^{-/-}$, TACI$^{-/-}$, or BCMA$^{-/-}$ adipocytes treated with saline (NS) or rBAFF (500 ng/ml). % increase relative to WT NS. **a** A single experiment, $n = 2$/condition. **b** Representative of three independent experiments, $n = 3$–6/condition. **c–d** Representative of three independent experiments, $n = 3$–6/condition. **e** Representative of two independent experiments, $n = 3$/condition. **f** Representative samples of a single experiment, $n = 3$. **g** Representative of two independent experiments, $n = 3$/condition. **b–e**, **g** For bar graphs, data represents mean±SEM. **b–e**, **g** Unpaired two-tailed Student's *t* test. *$p < 0.05$, **$p < 0.01$, ***$p < 0.001$. Source data are provided as a Source data file.

a large number (2535) of genes modulated by APRIL, as compared with BAFF in primary subcutaneous white adipocytes (Fig. 4c). APRIL specifically downregulated inflammatory pathways including IL6-mediated signaling, TLR signaling pathway, and inflammation-mediated by chemokine and cytokine signaling pathway (Supplemental Fig. 15a). Screening of obesity-associated proinflammatory cytokines and chemokines in subcutaneous white adipocytes demonstrated that BAFF, but not APRIL, upregulated the mRNA expression of multiple proinflammatory mediators, including *Tnf, Cxcl1, Ccl2, and Ccl3* (Supplemental Fig. 15b–e). Thus, these findings suggest that despite similar effects to modify adipocyte lipid handling, APRIL, unlike BAFF, may not induce an inflammatory response in subcutaneous white adipocytes.

**BAFF/APRIL axis is a contributor to diet-induced weight gain.** As APRIL modifies subcutaneous white adipocyte lipid handling, we next tested the effect of genetic deletion of APRIL on DIO. Surprisingly, APRIL$^{-/-}$ mice likewise exhibited augmented levels of systemic BAFF suggesting that deletion of BAFF or APRIL alone drives compensatory increases of the reciprocal family member (BAFF$^{-/-}$ Fig. 4b; APRIL$^{-/-}$; Fig. 5a). Consistent with the overarching observation of increased BAFF association with body weight control (Fig. 1e, f), APRIL$^{-/-}$ mice were resistant to DIO (Fig. 5b) with decreased pWAT and BAT weight (Fig. 5c), despite similar food intake (Fig. 5d), as compared with WT controls. This protection was associated with higher EE (Fig. 5e) and decreased metabolic sequelae, including glucose dysmetabolism (Fig. 5f–i) and hepatocellular damage (Fig. 5j).

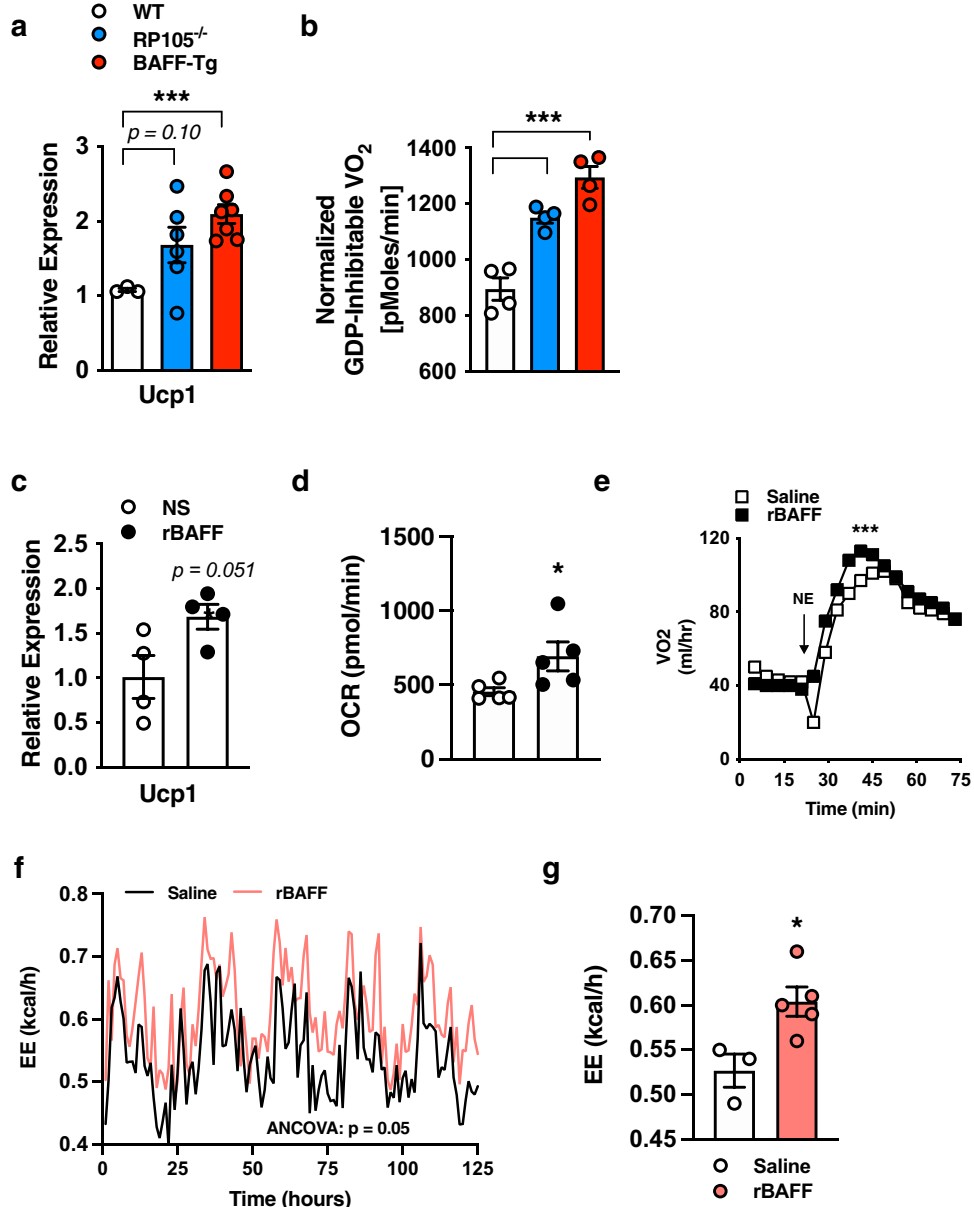

**Fig. 3 BAFF modifies brown adipose adaptive thermogenesis. a–b** WT, RP105$^{-/-}$, or BAFF-Tg mice were fed HFD for 20 weeks. **a** BAT *Ucp1* mRNA expression quantified by qPCR. **b** BAT mitochondria oxygen consumption. **c–d** Brown adipocytes were treated with saline (NS) or rBAFF (500 ng/ml) for 6 hours. **c** *Ucp1* mRNA expression quantified by qPCR, relative expression to NS. **d** Oxygen consumption rate (OCR). **e** Oxygen consumption of WT mice treated with saline (NS) or rBAFF (2 μg/mouse) for 24 hours prior to norepinephrine (NE; 1 mg/kg) challenge. **f–g** Obese WT mice treated with rBAFF (2 μg/mouse) every other day for 1 week and monitored in TSE Phenomaster. **f** Monitoring of energy expenditure over 5 days. **g** Bar graph of combined energy expenditure. **a–b** Representative of $n = 3–6$/condition. **c–d** Representative of three independent experiments, $n = 4–5$/condition. **e** Representative of two independent experiments, $n = 3–4$/condition. **f–g** A single experiment, $n = 3–5$/condition. **a–g** For bar and line graphs, data represents mean±SEM. **a–d** Unpaired two-tailed Student's *t* test. **e** Area under the curve. **f–g** analysis of covariance (ANCOVA) with body weight as covariate. *$p < 0.05$, ***$p < 0.001$. Source data are provided as a Source data file.

Given potential compensatory mechanisms between BAFF and APRIL, we next examined the contribution of the BAFF/APRIL axis to HFD-driven weight gain. Combined deletion of BAFF and APRIL in mice led to robust amplification of weight gain, above the levels of BAFF$^{-/-}$, APRIL$^{-/-}$, and WT mice (Fig. 6a). Further, BAFF$^{-/-}$/APRIL$^{-/-}$ mice fed HFD exhibited increased amounts of subcutaneous (iWAT) and visceral (eWAT, pWAT) AT (Fig. 6b). Lack of both BAFF and APRIL was associated with protection from glucose dysmetabolism (Fig. 6c–e) and a similar degree of hepatocellular damage (Fig. 6f).

We subsequently sought to initially interrogate the role of the BAFF/APRIL receptor(s) in HFD-driven weight gain. Mice lacking a single BAFF/APRIL receptor (BAFF-R$^{-/-}$, TACI$^{-/-}$, BCMA$^{-/-}$) were protected from weight gain (Supplementary Fig. 16a). The protection from weight gain was associated with increased systemic levels of BAFF (Supplementary Fig. 16b). Notably, only TACI$^{-/-}$ and BCMA$^{-/-}$ mice were protected from HFD-driven, glucose dysmetabolism, and hepatocellular damage compared with WT controls (Supplementary Fig. 16c–e). In contrast, BAFF-R$^{-/-}$ mice exhibited similar glucose dysmetabolism and hepatocellular damage compared with WT controls (Supplementary Fig. 16c–e).

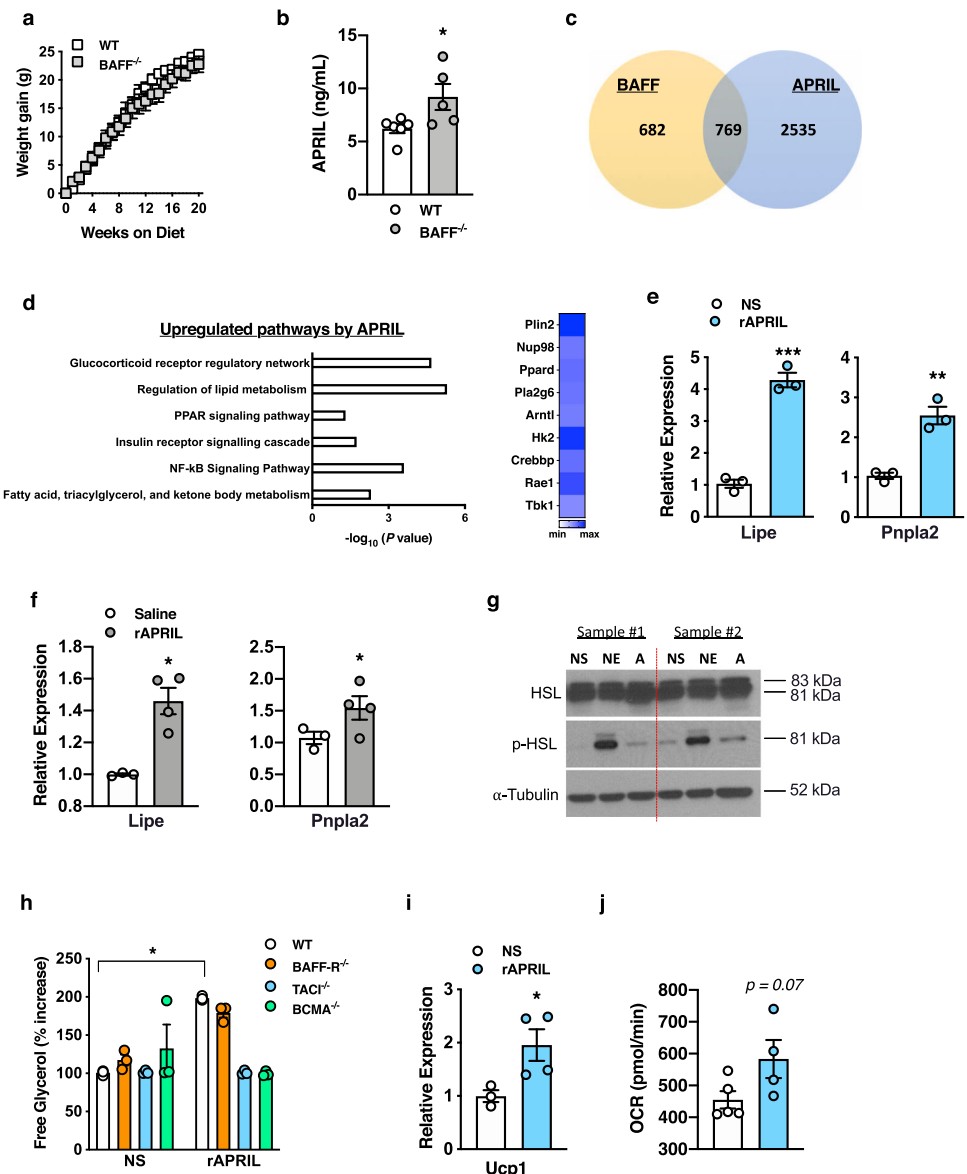

**Fig. 4 APRIL modifies white adipose lipolysis and brown adipose adaptive thermogenesis. a** Weight gain of WT or BAFF$^{-/-}$ mice fed HFD for 20 weeks. **b** Systemic APRIL concentration in WT or BAFF$^{-/-}$ mice quantified by ELISA. **c–d** White adipocytes stimulated in the presence or absence of rBAFF (500 ng/ml), rAPRIL (500 ng/ml), or saline and subjected to RNA-seq analysis. **c** Venn diagram analysis of genes modified by BAFF and APRIL. **d** Ontology pathways and heat map of associated overlapped genes. **e** qPCR quantified mRNA expression of indicated lipolysis genes in white adipocytes treated with saline (NS) or rAPRIL (500 ng/ml) for 24 hours. **f** Lean WT mice treated with rAPRIL (2 μg/mouse) every other day for 2 weeks. eWAT mRNA expression of indicated lipolysis genes quantified by qPCR, relative expression to saline-treated controls. **g** Primary white adipocytes from WT mice were treated with saline (NS), norepinephrine (NE; 1 μM) or rAPRIL (a; 500 ng/ml) for 12 hours. Protein expression of hormone-sensitive lipase (HSL), phosphorlayted HSL (p-HSL), and α-tubulin loading control by western blot. **h** Free glycerol in supernatants, quantified by colorimetric assay, of WT, BAFF-R$^{-/-}$, TACI$^{-/-}$, or BCMA$^{-/-}$ adipocytes treated with saline (NS) or rAPRIL (500 ng/ml). % increase relative to WT NS. **i–j** WT brown adipocytes treated with saline (NS) or rAPRIL (500 ng/ml) for 6 hours. **i** Ucp1 mRNA expression quantified by qPCR, relative expression to NS. **j** Oxygen consumption rate. **a–b** Representative of three independent experiments, n = 4–6/condition. **c–d** A single experiment, n = 2/condition. **e–f** Representative of three independent experiments, n = 3–4/condition. **g** Representative samples from a single experiment, n = 3/condition. **h** Representative of two independent experiments, n = 3/condition. **i** Representative of three independent experiments, n = 3–4/condition. **j** A single experiment, n = 4–5/condition. **a–b**, **e–i** For bar and line graphs, data represents mean±SEM. **a–b**, **e–i** Unpaired two-tailed Student's *t* test. *p < 0.05. Source data are provided as a Source data file.

Subcutaneous WAT from BAFF-R$^{-/-}$, TACI$^{-/-}$, or BCMA$^{-/-}$ mice exhibited similar mRNA expression of Ucp1-dependent thermogenic/beige markers (e.g., Pgc1a, Ucp1, Dio2, Cidea, Elovl6) as compared with WT counterparts (Supplementary Fig. 16f–j). Combined, these data indicate a role for the BAFF/APRIL axis in the development of obesity (Figs. 1h and 6a, Supplementary Fig. 16a).

**BAFF/APRIL axis effects are conserved in humans**. To determine whether BAFF and APRIL's effects were conserved in humans, primary human white adipocytes were isolated from omental WAT of a pediatric cohort of severely obese individuals undergoing surgical treatment (Supplementary Table 1). Consistent with our findings in mice, rBAFF and rAPRIL treatment resulted in enhanced human white adipocyte-free glycerol release

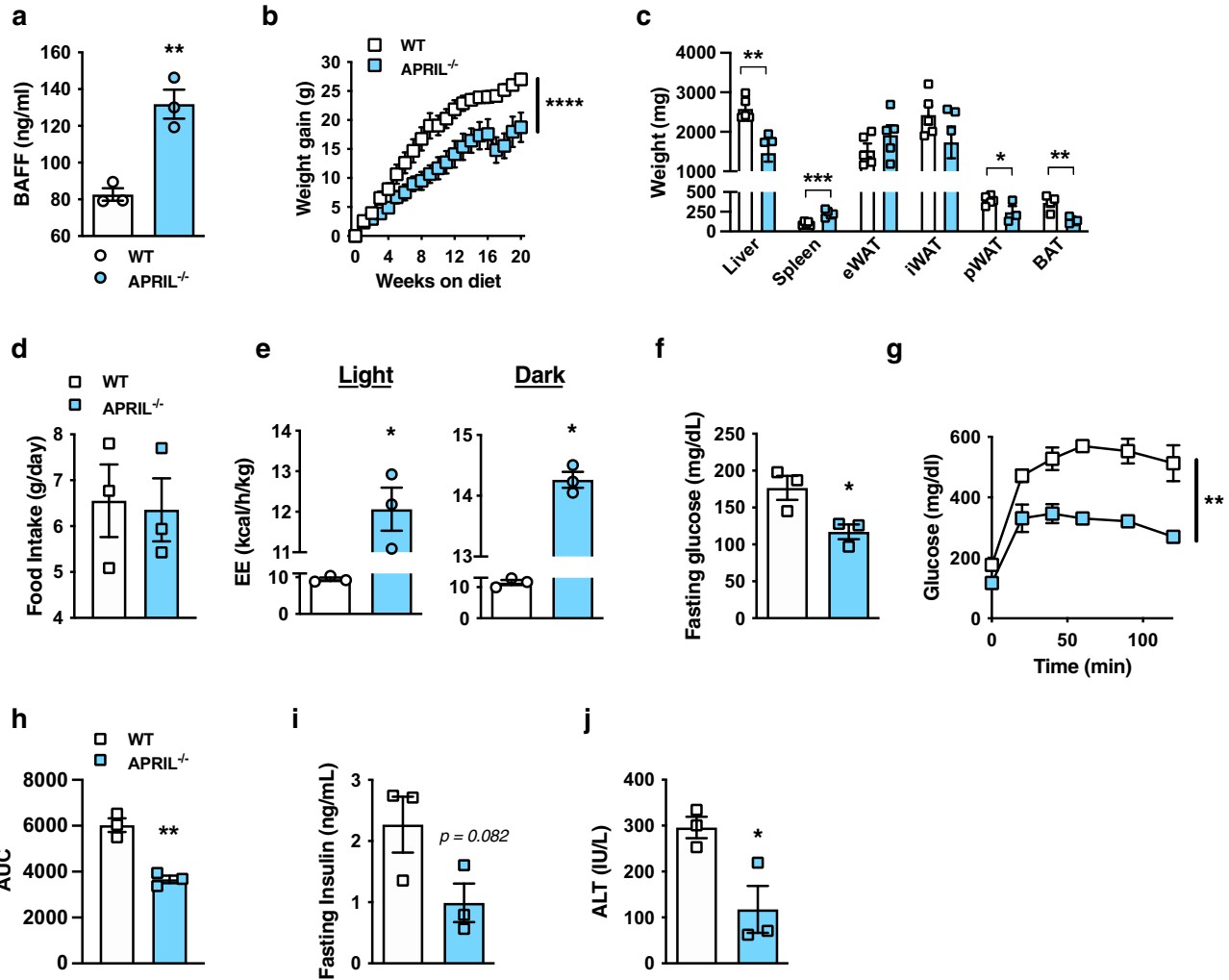

**Fig. 5 APRIL-deficient mice are protected from obesity development. a–j** WT and APRIL$^{-/-}$ mice were fed HFD for 20 weeks. **a** Systemic BAFF concentration quantified by ELISA. **b** Weight gain. **c** Tissue weight distribution as indicated at time of harvest. **d** Food Intake. **e** Combined energy expenditure at 16 weeks of HFD analyzed by TSE Phenomaster system. **f** Fasting glucose at 20 weeks of HFD. **g** GTT at 14 weeks of HFD. **h** AUC. **i** Fasting insulin at 20 weeks of HFD. **j** Systemic ALT at 20 weeks of HFD. **k** Weight gain of WT, BAFF$^{-/-}$, APRIL$^{-/-}$, or BAFF$^{-/-}$/APRIL$^{-/-}$ mice fed HFD for 20 weeks. **a–j** Representative of three independent experiments, n = 6-7/condition. **a–j** For bar and line graphs, data represent mean ± SEM. **a, c–f, i–j** Unpaired two-tailed Student's *t* test. *$p < 0.05$, **$p < 0.01$, ***$p < 0.001$. **b, g–h** Area under the curve. **$p < 0.01$, ****$p < 0.0001$. Source data are provided as a Source data file.

(Fig. 7a) and induced the mRNA expression of LIPE and PNPLA2 (Fig. 7b–c). At baseline, persons with severe obesity had lower levels of circulating BAFF but increased levels of APRIL as compared to lean controls (Fig. 7d, e; Supplementary Table 2). BAFF, but not APRIL, was negatively correlated with increased BMI (Fig. 7f, g). Notably, 1-year after bariatric surgery and after substantial weight loss (Supplementary Table 1), BAFF and APRIL levels were positively correlated with a greater decrease in BMI (Fig. 7h, i). Collectively, these findings suggest that the BAFF/APRIL axis potentially exerts conserved effects on human adipocytes and may act as a potential surrogate marker for the degree of weight loss after bariatric surgery.

## Discussion

The traditional perception of inflammation in the context of obesity has focused on the detrimental impact of inflammatory mediators on the pathogenesis of secondary metabolic derangements associated with obesity. Although inflammation undoubtedly exacerbates obesity-associated sequelae, the direct relevance to the regulation of weight gain remains underdefined. In this study, we

used a reductionist approach to tease apart the role of the BAFF/ APRIL axis in obesity. Our findings highlight the beneficial effects of the BAFF/APRIL axis in regulation of weight gain by altering AT metabolic function and energy homeostasis (Fig. 8). These observations add to the growing number of reports invoking a potentially dichotomous relationship presented by inflammation in obesity: (a) the potentially advantageous inflammatory contributors to obesity development[15,32,48] and (b) established obesity-driven inflammatory cytokines that can exacerbate downstream metabolic derangements[49]. Thus, a balance between these two functions of the immune system likely needs to be tightly controlled and regulated. We predict that cytokines, including BAFF and APRIL, can act to mobilize energy from AT in times requiring increased EE and lipid handling (e.g., obesity development, infections[50–54]). However, over-activation of these inflammatory pathways, including the BAFF/APRIL axis, can promote type 2 diabetes and autoimmune disorders (e.g., Systemic Lupus Erythematosus, Sjogren's Syndrome)[36]. Notably, involuntary weight loss[55] is a common sign accompanying some autoimmune diseases and an independent risk factor for worse outcomes[56,57].

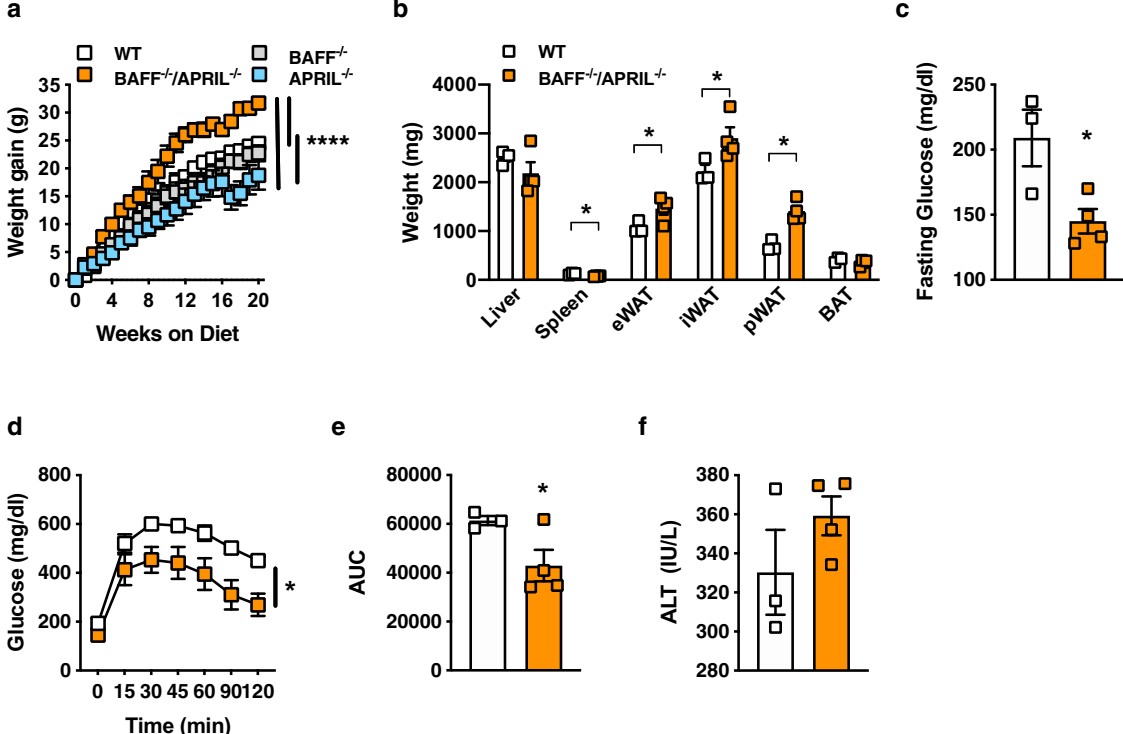

**Fig. 6 Lack of BAFF and APRIL exacerbates diet-induced weight gain. a** Weight gain of WT, BAFF$^{-/-}$, APRIL$^{-/-}$, and BAFF$^{-/-}$/APRIL$^{-/-}$ mice fed HFD for 20 weeks. **b–f** WT and BAFF$^{-/-}$/APRIL$^{-/-}$ mice fed HFD for 20 weeks. **b** Tissue weight distribution as indicated at time of harvest. **c** Fasting glucose at 14 weeks of HFD. **d** GTT at 14 weeks of HFD. **e** AUC. **f** Systemic ALT at time of harvest. **a** A single experiment, $n = 3–6$/condition. **b–f** A single experiment, $n = 3–4$/condition. **a–f** For bar and line graphs, data represent mean±SEM. **a**, **d** Area under the curve. **b–c**, **e–f** Unpaired two-tailed Student's $t$ test. **a–e** $*p < 0.05$, $****p < 0.0001$. Source data are provided as a Source data file.

In agreement with our findings (Supplemental Fig. 11b), published studies harnessing BAFF$^{-/-}$ mice also revealed increases in body weight and weight gain[44,46,47]. Published evidence suggests that BAFF may modulate obesity-associated glucose dysmetabolism in a sex-dependent manner[44]. The potential impact of this finding is bolstered by the observed sexual dimorphism in normal human BAFF levels[58]. Traditionally, female mice, unless genetically (ob/ob; db/db)[59] or hormonally[60] manipulated, are protected from severe DIO and consequent pathogenesis of obesity-associated sequelae. Congruently, female cohorts in these studies gained ~1 g of body weight on an HFD (total body weight of 19 g after 4 weeks on a HFD)[44]. The limitations of this approach make it difficult to draw robust conclusions regarding gender-specific mechanisms. Thus, future in-depth interrogation of how the BAFF/APRIL axis impacts obesity-associated glucose dysmetabolism, the relevance of gender-specific mechanisms would be warranted, possibly through the exploitation of the thermoneutral housing model[61]. Likewise, the impact of aging, a well-appreciated modifier of AT homeostasis and biology, on gender-specific mechanisms and BAFF/APRIL axis modulation of weight gain would be highly pertinent.

BAFF and APRIL are traditionally perceived as regulators of B-cell maturation and survival. BAFF impacts the effector capacity of multiple immune cells (e.g., macrophages, B cells, T cells)[36]. Various immune cells infiltrate AT in obesity[62]. We hypothesize that this interplay between BAFF/APRIL, immune cells, and AT is likely an important contributor to the overall AT biology and function. Hence, detailed interrogation of how BAFF/APRIL axis-driven modulation of immune cells shapes adipocyte biology and weight gain represents key future studies. B cells infiltrate AT in the context of obesity and have been suggested to contribute to

the downstream metabolic derangements of obesity[63]. Our data indicate that multiple lines of mice with increased systemic BAFF, including RP105$^{-/-}$ or BAFF-Tg (augmented B-cell numbers) and μMT, APRIL$^{-/-}$, BAFF-R$^{-/-}$, TACI$^{-/-}$, or BCMA$^{-/-}$ mice (devoid or decreased number of mature B cells) are similarly protected from DIO and the subsequent downstream metabolic sequelae (Fig. 1, Fig. 5, Supplementary Fig. 16). The consistent unifying factor among these different transgenic lines is elevated systemic BAFF levels, invoking that observed BAFF effects in our studies may be independent of B cells. In contrast to our findings, existing reports indicate that μMT mice gain similar weight[63] to WT counterparts. Nonetheless, reported protection from obesity-associated glucose dysmetabolism[63] is congruent with our findings. As B cells contribute to the establishment of the gut microbiota (e.g., IgA regulation[64]), it is plausible that lack of B cells directs differential gut microbial colonization within μMT mice harbored at different facilities.

The microbiome modulates body weight[65]. Our examination of the microbiome in one model of increased circulating BAFF, RP105$^{-/-}$ mice, revealed a similar, HFD-driven, microbiome in WT and RP105$^{-/-}$ mice (Supplementary Fig. 1f). These findings would suggest that the microbiome may not play an integral role in the reversal of weight gain in RP105$^{-/-}$/BAFF$^{-/-}$ mice. However, these findings could be limited to the RP105$^{-/-}$ system. We posit that analyses of multiple mouse lines with varying levels of systemic BAFF (RP105$^{-/-}$, RP105$^{-/-}$/BAFF$^{-/-}$, BAFF$^{-/-}$, μMT, BAFF-Tg, APRIL$^{-/-}$, BAFF$^{-/-}$/APRIL$^{-/-}$, BAFF-R$^{-/-}$, TACI$^{-/-}$, BCMA$^{-/-}$), would likely reveal differential intestinal microbiome composition. However, the possible differences may not correlate with different BAFF levels in these mice, but rather may be dependent on direct alteration of key biological processes in these specific settings such as BAFF-driven modulation of

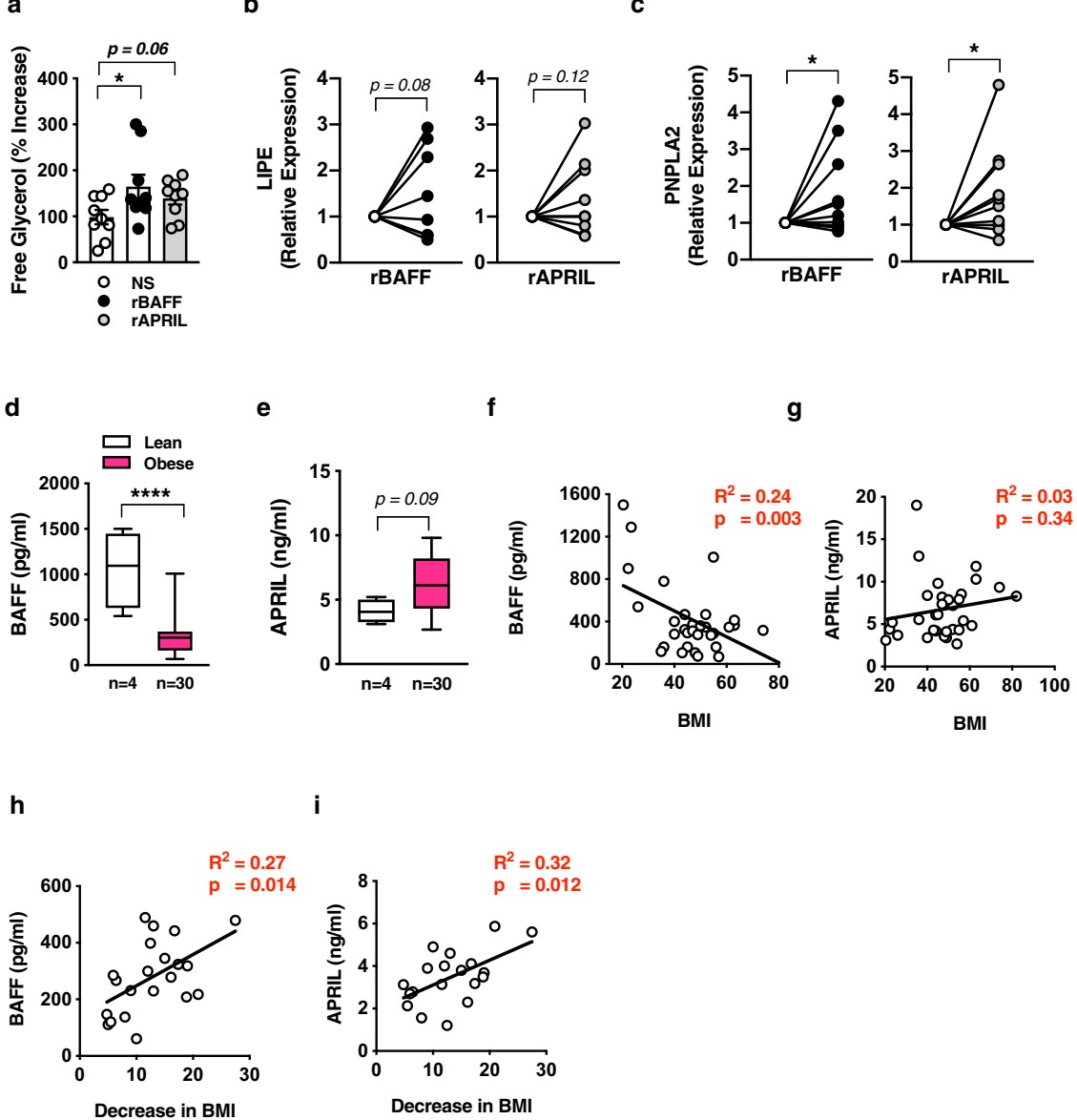

**Fig. 7 Effect of BAFF and APRIL axis is conserved in human adipocytes. a–b** Human primary adipocytes isolated from persons undergoing bariatric surgery. Adipocytes were treated with saline (NS), rBAFF (500 ng/ml), or rAPRIL (500 ng/ml) for 24 hours as indicated. **a** Supernatant free glycerol quantified by colorimetric assay. **b** LIPE mRNA expression quantified by qPCR, normalized to NS. **c** PNPLA2 mRNA expression quantified by qPCR, normalized to NS. **d** BAFF and **e** APRIL systemic concentration in lean or obese persons quantified by multiplex. **f** BAFF and **g** APRIL systemic concentration correlation with BMI. **h** BAFF and **i** APRIL systemic concentration correlation with decrease in BMI 1-year after bariatric surgery. **a–c** Representative persons, $n = 9$–$10$/condition. **d–g** Representative patients, $n = 4$ lean and $n = 30$ obese. **h–i** Data combined, $n = 22$ persons undergoing bariatric surgery. **a–c** For bar and line graphs, data represents mean±SEM. **d–e** For box plots, the midline represents the mean, boxes represent the interquartile range and whiskers show the full range of values. **a–e** Unpaired two-tailed Student's $t$ test. **a, c, d** *$p < 0.05$, ****$p < 0.0001$. **f–i** Linear regression. Source data are provided as a Source data file.

$T_{regs}$[11]. Further, BAFF-Tg mice exhibit enhanced, commensal-flora dependent, systemic IgA levels.[66] Whether and how systemic IgA and presence of plasma cells in the gut, in the setting of the BAFF axis, affects intestinal homeostasis and consequently contribute to obesity and weight gain remains unknown. Future, in-depth interrogations including, but not limited to, microbiome depletion, microbiome transfer studies, and analysis of these mice under germ-free conditions would provide further clarity to our understanding of the microbiome within the BAFF/APRIL axis in the regulation of body weight.

Despite similar capacities of BAFF and APRIL to modify subcutaneous white and brown adipocyte lipid handling, we demonstrate that BAFF$^{-/-}$ or APRIL$^{-/-}$ mice do not exhibit increased obesogenic diet-driven weight gain (Figs. 4a and 5b). However, BAFF$^{-/-}$ mice exhibit ~1.5-fold (48%) increase in APRIL, whereas APRIL$^{-/-}$ mice similarly exhibit approximately 2-fold (59%) increase in systemic BAFF (Figs. 4b and 5a)—likely stemming from the activation of compensatory mechanisms. Our findings that lean individuals have higher levels of BAFF than obese individuals, but lower APRIL levels, suggest possible reciprocal regulation between BAFF and APRIL. Inflammatory mediators within the same family (e.g., IL-17a and IL-17f)[67] can regulate each other's expression. Thus, examination of the interlink between BAFF and APRIL expression is warranted. As

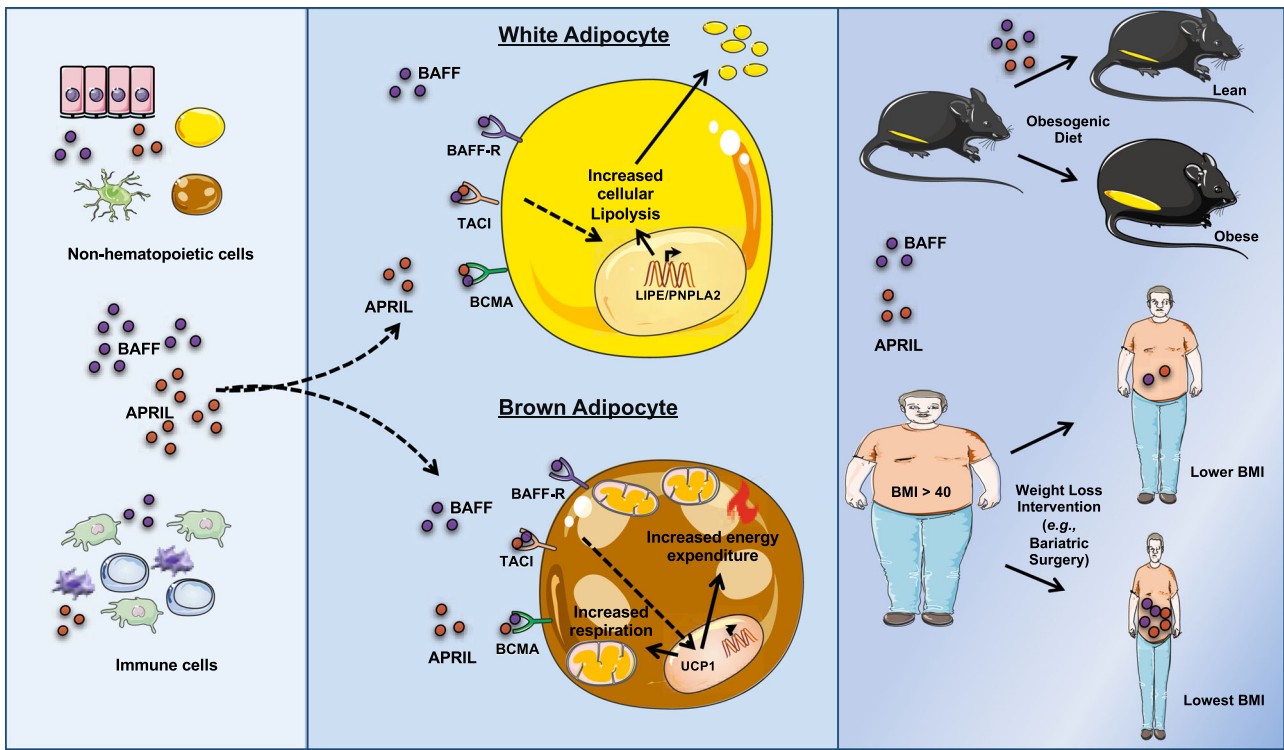

**Fig. 8 BAFF and APRIL axis regulates adipose tissue/adipocyte function and body weight gain.** A proposed model of the impact of the BAFF/APRIL axis on white and brown AT/adipocyte lipid handling. BAFF/APRIL regulation of AT/adipocytes function is associated with protection from DIO in mice. The effect of this axis positively correlates with greater loss of body weight in bariatric patients.

lack of both BAFF and APRIL brought out robust DIO-driven weight gain (~50% increase Fig. 6a), this suggests that the BAFF/APRIL system likely works in concert to regulate weight gain.

Recent reports suggest that BAFF and APRIL's receptors, including BAFF-R and TACI, regulate obesity pathogenesis[40–43]. Although BAFF-R$^{-/-}$ or TACI$^{-/-}$ mice gain significantly more weight on HFD[40–43], these mice are protected from the downstream glucose dysmetabolism. Our data indicate that mice with a lack of a single receptor (BAFF-R$^{-/-}$, TACI$^{-/-}$, BCMA$^{-/-}$), regardless of which, are all protected from HFD-driven weight gain (Supplementary Fig. 16a). Similarly, we also show that deficiency of a single receptor is sufficient to induce increased systemic levels of BAFF (Supplementary Fig. 16b). In agreement with published reports, our findings also demonstrate that BAFF-R$^{-/-}$[43], TACI$^{-/-}$[41] mice are protected from HFD-driven weight and TACI$^{-/-}$ mice are protected from glucose dysmetabolism[41]. Conversely to our data, existing evidence suggests that BAFF-R$^{-/-}$ mice are protected from glucose dysmetabolism but exhibit enhanced NAFLD pathogenesis[43] and enhanced steatosis[43]. As BAT activity is linked with the development of steatosis[68–70], it is plausible that augmented steatosis in BAFF-R$^{-/-}$ mice consist of dampened BAT activity or even direct effect of BAFF on hepatocytes via BAFF-R signaling. To our knowledge, our data describing the role of BCMA in HFD-driven obesity represents the first depiction on this topic. Given the intricate complexity of this system, in-depth exploration of the receptor side of the BAFF/APRIL axis is clearly warranted. Further, targeted tissue/cell type manipulation of BAFF/APRIL axis specific receptor(s) is needed and could be potentially exploited as effective preventative and/or therapeutic method for modifying weight gain.

Our findings indicate that BAFF-R impacts BAFF-driven subcutaneous white adipocyte lipolysis, whereas TACI and BCMA modulate BAFF and APRIL-induced lipolysis—findings in line with the selectivity in BAFF and APRIL binding to these receptors (Figs. 2g and 4h). As a single receptor deletion (BAFF-R$^{-/-}$, TACI$^{-/-}$, BCMA$^{-/-}$), deletion of BAFF-R/BCMA (Supplementary Fig. 7 and 13), or abolishment of all three receptors (Supplementary Fig. 7 and 13) abrogates BAFF or APRIL-driven lipolysis, this suggests the possibility that engagement of multiple receptors on subcutaneous white adipocytes is necessary to provide sufficient signal strength for activation of lipolysis. Hence, closer examination of the critical BAFF/APRIL receptor(s) involved with lipolysis, thermogenesis, and its capacity to control body weight would be of significant future interest.

BAFF/APRIL effects on lipolysis are distinctive in subcutaneous white adipocytes, whereas induction of *Ucp1* expression is specific to brown adipocytes (Supplementary Fig. 8). Consistent with this, subcutaneous WAT isolated from mice with increased BAFF exhibited similar mRNA expression of brown/beige thermogenic markers to WT counterparts (Supplementary Fig. 9, Supplementary Fig. 16f). Combined our findings hint that the BAFF axis may be insufficient for subcutaneous white adipocyte browning. However, the full definition of the BAFF/APRIL axis' ability to modify Ucp1-dependent and Ucp1-independent browning/beiging of subcutaneous white adipocyte is of strong interest. Specifically, like the dependency of Ucp1 deletion on weight gain is uncovered at thermoneutrality[71], future examination of BAFF/APRIL axis at thermoneutrality might be warranted. This layered complexity of the BAFF/APRIL system (two molecules and three receptors) invokes the possibility of a highly regulated mechanism(s) to impact function across different tissues/cell types (e.g., WAT, BAT, liver). In addition, although our findings suggest responsiveness by WAT and BAT to BAFF and APRIL, the relevant source of BAFF and/or APRIL in the context of an obesogenic diet remains unknown. Both hematopoietic and non-hematopoietic cells, including adipocytes, can produce BAFF and APRIL[36,37,72]. Complicating matters further, BAFF and

APRIL exist as oligomers, homo/hetero multimers and their conformation regulate the capacity of BAFF and APRIL to activate their three receptors[73]. Thus, future exploration of the conformation and relevant sources of BAFF and APRIL in the context of obesity and control of body weight should be investigated.

BAFF and APRIL have the capacity to modify DIO in mice (Figs. 1h, 5b, and 6a). In addition, the effects of both BAFF and APRIL are conserved in primary human adipocytes. Novel preliminary findings of a direct correlation between higher BAFF and APRIL levels with greater changes in BMI after bariatric surgery indicate that the BAFF and/or APRIL axis may play a role in human weight regulation after bariatric surgery and could represent a prognostic marker for weight loss outcomes. However, our present study is limited by a pediatric cohort and comparison of our obese cohort with a small number of lean individuals. Thus, such limitations directly invoke the need for future expanded and thorough exploration of the BAFF/APRIL axis in humans to generate the most robust conclusions regarding the impact of these cytokines on human weight regulation after bariatric surgery. Recent GWAS studies have revealed the association of variants within *TNFSF13B* (the gene encoding BAFF) and near *TNFSF13* (the gene encoding APRIL) with lipid metabolism (e.g., HDL-C).[74] As subcutaneous AT lipolysis is an independent contributor to inter-individual variation in HDL-C[75], these findings further support the applicability of our mechanistic murine data to understanding how these cytokines regulate metabolism and adiposity. In this regard, even a minimal increase in BAFF (~20% increase in RP105$^{-/-}$ mice) are protective of DIO and would likely not cause the deleterious effects observed with log-increases of BAFF[76,77] (e.g., autoimmunity, B-cell-driven pathology) invoking therapeutic potential in humans. In contrast to BAFF, APRIL does not appear to induce a proinflammatory program in adipocytes (Supplementary Fig. 15) while maintaining its capacity to modulate white and brown adipocyte lipid handling and play a role in protection from DIO (Figs. 4–7). Thus, an approach harnessing the BAFF/APRIL axis may need careful monitoring of the levels of BAFF/APRIL in order to provide a maximal therapeutic index.

In summary (Fig. 8), our data demonstrate the impact of the BAFF and APRIL axis as beneficial mediators of white and brown adipocyte lipid handling and body weight control. These observations may highlight a prospective and appealing pathway open to the development of new clinical approaches for body weight modulation.

## Methods

**Reagents**. All cell culture reagents were endotoxin-free to the limit of detection of the Pierce Chromogenic Endotoxin Quantification Assay. Recombinant BAFF was also heat-inactivated via boiling for 30 min and used as a control for free glycerol release studies.

**Mouse obesogenic diet model**. All experiments utilized male mice, 6–8 weeks old, on a C57BL/6 background. WT (Jackson), RP105$^{-/-}$ (in house)[20], RP105$^{flox}$ (in house), Vav1$^{cre}$ (Jackson), CD19$^{cre}$ (Jackson), Nestin$^{cre}$ (Jackson), MLC$^{cre}$ (generously provided by Steven Burden[78], RP105$^{-/-}$/BAFF$^{-/-}$ (in house), BAFF$^{-/-}$ (Jackson), μMT (Jackson), BAFF-Tg (Biogen Idec), APRIL$^{-/-}$ (Jackson), BAFF-R$^{-/-}$ (generously provided by Klaus Rajewsky[79]), TACI$^{-/-}$ (generously provided by Richard Bram[80]), BCMA$^{-/-}$ (generously provided by Loren Erickson[81], BAFF-R$^{-/-}$/BCMA$^{-/-}$ (in house), BAFF-R$^{-/-}$/TACI$^{-/-}$/BCMA$^{-/-}$ (in house), and BAFF$^{-/-}$/APRIL$^{-/-}$ (in house) mice were bred at Cincinnati Children's Hospital Medical Center (CCHMC) in a specific pathogen-free (spf) facility maintained at 22℃, with free access to autoclaved chow diet food (CD; LAB Diet #5010; calories provided by carbohydrates [58%], fat [13%] and protein [29%]) and water. Mice, 6–8 weeks of age, were fed either an irradiated HFD (Research Diets #D12492i; 60% of calories from fat) or a CD and food was replaced weekly. For energy absorption studies, fecal pellets were collected at 15 weeks and analyzed by bomb calorimetry. For lipid absorption studies, at 24 weeks, mice were switched to Behenate mouse diet and fecal pellets were collected at day 3 and day 4 for fat absorption analysis[82,83]. Phylogenetic analysis of the fecal microbiota was performed essentially as previously described via 16 S rRNA pyrosequencing of fecal pellets collected after 8 weeks of HFD feeding[84]. Body composition and in vivo metabolic phenotyping were performed as previously described[61,85–87]. Prior to glucose metabolism testing or terminal harvest, mice were fasted overnight as previously described[61,85–87]. For glucose tolerance tests, fasted mice were subjected to glucose (mice injected 100 μL of a 10% dextrose solution per gram of body weight) and glucose was monitored kinetically using an Accu-Chek glucometer per manufacturer instructions. Systemic BAFF (R&D systems) and APRIL (MyBioSource) levels were quantified by ELISA as per manufacturer instructions. All care was provided in accordance with the Guide for the Care and Use of Laboratory Animals. All studies were approved by the CCHMC IACUC.

**Mouse indirect calorimetry and EE**. EE, via indirect calorimetry, was quantified by Phenomaster (TSE systems)[86,88–91]. Gases ($O_2$ and $CO_2$) in the metabolic chambers were calibrated and equilibrated prior to the initiation of the study according to manufacturer instructions. Mice were subsequently housed in individual boxes and acclimated for 2 days before the start of the study. 12-hour light and dark cycles were maintained throughout. Data points were continuously collected for a total of 5–8 days. Gas exchange ($O_2$ and $CO_2$) and locomotor activity were recorded every 15 minutes and EE was calculated according to the manufacturer guidelines. Ambient temperature (22℃) and humidity were maintained via climate-control units in the metabolic chambers. For exogenous rBAFF or rAPRIL studies, rBAFF (2 μg/mouse), rAPRIL (2 μg/mouse), or saline was administered via intraperitoneal injection every 48 hours.

**Bone marrow transfer**. Bone marrow from WT were transferred into WT or RP105$^{-/-}$ recipient mice as previously described[86,92]. In brief, femur-derived bone marrow cells ($5 \times 10^6$) were transferred into whole-body irradiated RP105$^{-/-}$ recipient mice. Flow cytometric analysis of peripheral blood chimerism was performed 10 weeks after bone marrow reconstitution prior to mice being placed on an HFD or CD.

**Generation of RP105Flox/Flox line**. An ES Cell clone (D11) containing the conditionally targeted allele, Cd180$^{tm1a(KOMP)Wtsi}$, of Cd180, the gene encoding RP105, was obtained from the knockout mouse project (KOMP). This clone passed quality control and was selected for blastocyst injection. ES cells were grown on neomycin-resistant mouse embryonic fibroblast feeders (a gift of M. Flick) and were then injected into C57BL/6–Albino blastocysts (B6(Cg)-Tyr$^{c-2J}$/J stock #00058 from Jackson Labs http://jaxmice.jax.org/strain/000058.html) and implanted into pseudopregnant C57BL/6 mothers by the CCHMC transgenic mouse core. These embryo transfers yielded numerous chimeric males of high chimerism, several of which were capable of transmitting the targeted allele in the germline as confirmed by PCR.

The germline-targeted progeny of these chimeras were subsequently bred to FLPe-Deleter mice[93] (B6.Cg-Tg(ACTFLPe)$^{9205Dym}$/J stock #005703 from Jackson Labs http://jaxmice.jax.org/strain/005703.html) to excise the FRT-flanked knockout-first β-galactosidase reporter/selection cassette[94]. Excision of the cassette was confirmed by PCR for both the neomycin selection cassette and with primers (IH152 and IH154) flanking the FRT cassette. This PCR product was sequenced to confirm the presence of both the FRT and loxP sites. For colony maintenance, the presence of the Exon 3 distal loxP site was confirmed by PCR using primers complementary to the genomic DNA flanking the loxP site (IH123 and IH124). This product was also sequenced to confirm the presence of the loxP site. The Rp105$^{Flox/WT}$ mice thus generated were then bred to C57BL/6NJ mice. Progeny lacking the FlpE-deleter transgene (confirmed via PCR) were selected to establish an Rp105$^{Flox/Flox}$ breeding colony after breeding to homozygosity for both the Floxed allele and a wild-type *Nnt* allele[95]. Genotyping of RP105$^{Flox/Flox}$ mice was carried out with the following primer pairs: FWD: *CCATCTGAGAAA GAAGAGCATTTACC*; REV: *TGAGCATTAGATTTTGCTGGGAC*; and FWD: *GCATTTCCCCATCTATCATCTGAC*; REV: *TATTGCTAACATCGTCCGCCTAC*. Exon 3 containing the majority of the coding sequence of *RP105 (Cd180)*, was identified as a likely to result in knockout by a prediction algorithm used by KOMP and was successfully targeted in the generation of the germline RP105 knockouts[96].

**Mouse primary subcutaneous and brown adipocytes**. Isolation and digestion (1 mg/ml Collagenase Type IV, Dispase 2, $CaCl_2$) of subcutaneous white AT[97] (iWAT). Preadipocytes within stromal vascular fraction (SVF) were cultured until confluence and differentiated as previously described[97]. In brief, initiation media (Growth media [Dulbecco's Modified Eagle Medium (DMEM):F12, fetal bovine serum (FBS), Pen/Strep], Rosiglitazone, Dexamethasone, 3-Isobutyl-1-methyl-xanthine, insulin) was utilized for 2 days, afterward, cells were changed to continuation media (Growth media, Rosiglitazone, Insulin) for 2 days and followed by differentiation media (Growth media, Insulin) for an additional 2 days. Differentiated white adipocytes were utilized for downstream processes.

Brown preadipocytes were isolated from P1 B6/J neonates, maintained in high-glucose DMEM in incubators at 37℃ and 5% $CO_2$ and immortalized with pBabe-SV40 Large T.[98] The gender of neonates was not determined. Brown preadipocytes were subjected to differentiation media (high-glucose DMEM including 10% FBS, 1% antibiotics, 20 nM insulin and 1 nM $T_3$). After 4 days, cells were switched to

induction media (high-glucose DMEM including 10% FBS, 1% antibiotics, insulin 20 nM, 1 nM T₃, 0.125 mM indomethacin, 2 μg/mL dexamethasone and 0.5 mM 3-isobutyl-1-methylxanthine (IBMX) for 2 days. Following this, differentiation media was utilized until maximal differentiation on day 12. Maximally differentiated brown adipocytes were utilized for subsequent downstream studies.

**Adipocyte-free glycerol quantification.** Murine or human primary adipocytes were cultured in the presence or absence of rBAFF (500 ng/ml; R&D Systems) or rAPRIL (500 ng/ml; R&D Systems) for 24 hours. Free glycerol was quantified by colorimetric assay (Sigma Aldrich) as per manufacturer's instructions.

**qRT-PCR.** AT and adipocytes were homogenized in TRIzol (Invitrogen) followed by RNA extraction, reverse transcription to cDNA (Verso cDNA Synthesis Kit; Thermo Scientific) and qPCR analysis (Light Cycler 480 II; Roche)—according to manufacturer's instruction[61,86,87,92,99].

The following primer pairs were used for mouse studies: *Bactin* For GGCC CAGAGCAAGAGAGGTA Rev GGTTGGCCTTAGGTTTCAGG—*Lipe* For TCT CGTTGCGTTTGTAGTGC Rev ACGCTACACAAAGGCTGCTT—*Pnpla2* For GTTGAAGGAGGGATGCAGAG Rev GCCACTCACATCTACGGAGC—*Ucp1* For TCAGCTGTTCAAAGCACACA Rev GTACCAAGCTGTGCGATGTC—*F4/80* For CTTTGGCTATGGGCTTCCAGTC Rev GCAAGGAGGACAGAGTTTAT CGTG—*Cd68* For CTTCCCACAGGCAGCACAG Rev AATGATGAGAGGCAGC AAGAGG—*Tnf* For CCAGACCCTCACACTCAGATCA Rev CACTTGGTGGTT TGCTACGAC—*Il1b* For GGTCAAAGGTTTGGAAGCAG Rev TGTGAAATGC CACCTTTTGA—*Ifny* For TGGCTGTTTCTGGCTGTTACTG Rev ACGCTTAT GTTGTTGCTGATGG—*Ccl2* For TGTCTGGACCCATTCCTTCTTG Rev AGAT GCAGTTAACGCCCCAC—*Ccl3* For ACCATGACACTCTGCAACCAAG Rev TT GGAGTCAGCGCAGATCTG—*Il6* For TGGTACTCCAGAAGACCAGAGG Rev AACGATGATGCACTTGCAGA—*Cxcl1* For ACCCAAACCGAAGTCATAGC Rev TCTCCGTTACTTGGGGACAC—*Pgc1a* For CCCTGCCATTGTTAAGACC Rev TGCTGCTGTTCCTGTTTTC—*Dio2* For CAGTGTGGTGCACGTCTCCAA TC Rev TGAACCAAAGTTGACCACCAG—*Cidea* For TGCTCTTCTGTATCG CCCAGT Rev GCCGTGTTAAGGAATCTGCTG—Elovl6 For TCAGCAAAGCA CCCGAAC Rev AGCGACCATGTCTTTGTAGGAG. mRNA expression of each gene was compared to *Bactin* (beta-actin, mouse) expression. The following primer pairs were used for human studies: *hUBIQ* For CACTTGGTCCTGCGCTTGA Rev CAATTGGGGAATGCAACAACTTTAT—*hLIPE* For GAAGGCTATGTTGTCCT CCG Rev ATGAGAAAACCAGTGCTCGG—*hPNPLA2* For ACCTCAATGAAC TTGGCACC Rev CAACGCCACGCACATCTA. mRNA expression of each gene was compared to *hUBIQ* (ubiquitin, human) expression.

**RNA-sequencing and gene expression quantification.** Gene expression of primary white adipocytes was determined[100] by running 50 base pair single-end reads (~30 million reads per sample). Following the removal of barcodes and primers, raw reads were aligned to the mm10 genome using kallisto[101], which quantifies transcript abundances of high-throughput sequencing reads. Kallisto pseudoaligns reads to a reference, by identifying transcripts that are compatible with each raw read, with annotations provided by UCSC. The psuedoalignment generates accurate quantification, with transcripts per million (TPM) as the output. All further processing and analyses were performed in GeneSpring 14.9 GX. Each transcript was log2-normalized and baselined to the median across all samples. Reasonably expressed transcripts (raw TPM > 3 in 100% of samples in at least one condition) were included for differential analysis. Differential expression was determined through unpaired *t* tests with an false discovery rate-corrected *p* value cutoff of 0.05 and a fold change requirement of >2. For pathway analysis, the database at toppgene.cchmc.org was employed, which amasses ontological data from over 30 individual repositories. RNA-sequencing raw data can be accessed at GSE131298.

**Subcutaneous white adipocyte western blot.** Differentiated primary white adipocytes were stimulated with saline (NS), rBAFF (500 ng/ml), or norepinephrine (1 μM) for 12 hours and cells were then lysed in a buffer containing 50 mM Hepes, pH 7.4, 120 mM NaCl, 2 mM EDTA, 10 mM sodium pyrophosphate, 10 mM sodium β-glycerophosphate 10% CHAPS and protease & phosphatase inhibitors.[98] Total protein was quantified by Bradford protein assay and equivalent amounts of total protein from each treatment group was separated using acrylamide/bis-acrylamide gels and transferred to polyvinylidene difluoride membranes. Membranes were incubated for overnight with primary antibodies including α-tubulin (Cell signaling technology Cat #2125) diluted in 5% milk/phosphate-buffered saline with Tween® 20 (PBST) (1:1000), HSL (Cell signaling technology Cat #18381) diluted in 5% milk/PBST (1:1000) or phosphorylated-HSL (p-HSL; Cell signaling technology Cat #4126) diluted in (5% BSA/PBST). Thereafter, the membranes were incubated for one hour with specific horseradish peroxidase-conjugated secondary antibodies. Western blots were developed by enhanced chemiluminescence (PerkinElmer) per manufacturer instructions and detected by X-ray films.

**Cellular bioenergetics quantification.** Primary brown adipocytes (1 × 10⁴/well) or BAT-derived mitochondria were plated in a polyethylenimine pre-coated XF24 Cell culture microplate[86]. An XF Analyzer (Seahorse Bioscience) was used to measure bioenergetics. An XF24 extracellular flux assay cartridge (Seahorse

Bioscience) was hydrated overnight at 37°C according to manufacturer's instruction. MAS-1 buffer supplemented with sucrose (70 mM), D-mannitol (200 mM), KH₂PO₄ (5 mM), Mg₂Cl (5 mM), HEPES (2 mM), EDTA (1 mM) and 0.2% fatty free acid BSA (pH 7.4) was incubated at 37°C in a non-CO₂ incubator for 1 hour. Pyruvate/malate (30 mM), GDP (10 mM), FCCP (40 μM), antimycin (40 μM) were sequentially injected and cellular oxygen consumption rate (OCR) was quantified.

**Human subjects.** Lean or bariatric surgery participants were recruited and informed consent obtained from participants from the Cincinnati Children's Hospital Medical Center (CCHMC) Pediatric Diabetes and Obesity Center. Exclusion criteria included alcohol abuse, viral and autoimmune hepatitis, immunosuppressive or steroid use. Clinical phenotypes for lean or individuals with severe obesity at time of surgery and one-year post-surgery is provided in Supplemental Table 1 and 2. Recruitment and study protocols were approved by the institutional review board at CCHMC and studies were carried out in accordance with these guidelines.

**Human primary adipocytes.** Isolation and digestion of omental WAT (40 mg/ml Type II Collagenase) collected at the time of surgery[102]. Filtration of digested tissue was subsequently centrifuged at 250 g and subjected to Ack lysis buffer to isolate the SVF. SVF was cultured in expansion media (DMEM/F:12, 15% FBS, 1% Pen–strep) until confluence and subjected to human adipocyte differentiation media (DMEM/F:12, 1% Pen–strep, 2 mM glutamine, 15 mM HEPES, 10 mg/ml transferrin, 33 μM biotin, 0.5 μM insulin, 17 μM pantothenate, 0.1 μM dexamethasone, 2 nM T3, 500 μM IBMX, 1 μM ciglitazone) for 14-–16d, followed by 7–10d in human adipocyte maintenance media (DMEM/F:12, 1% Pen–strep, 2 mM glutamine, 15 mM HEPES, 10 mg/ml transferrin, 33 μM biotin, 0.5 μM insulin). Differentiated adipocytes were utilized for downstream processes.

**Human systemic cytokine quantification.** Human BAFF or APRIL plasma concentrations from lean or persons with severe obesity were determined by ELISA using Milliplex™ Multiplex kits (MilliporeSigma) according to manufacturer's protocol. In brief, 25 μL of plasma, plated in duplicate on a 96-well black plate, was incubated with 25 μL of antibody-coated beads. Plates were subsequently washed and 25 μL of secondary antibody was incubated, then by 25 μL of streptavidin-RPE. In all, 150 μL of sheath fluid was added to plates that were washed and then read using luminex technology on a Milliplex Analyzer (milliporeSigma). Data analysis performed by the Cincinnati Children's Medical Center Research Flow Cytometry Core.

**Statistical analysis.** Statistical tests were utilized for all data sets with similar variance. Choice of test was dependent on number of groups and whether normal distribution exists. For all normally distributed data Student's *t* test was used for two groups, whereas one-way analysis of variance was utilized for three or more groups with Tukey's post hoc test to determine differences between groups. Collective indirect calorimetry data was analyzed using analysis of covariance with body weight as covariates.[103] All data presented as means±SEM. *P* values < 0.05 were considered significant. Analysis was performed via GraphPad Prism Software. Determined sample sizes were based on preliminary data with respect to obesity modeling including weight gain, immune cell infiltration, severity of obesity-associated sequelae, and interrogation of myeloid cell inflammatory vigor. No animals were excluded from the analyses and none of the studies were blinded.

**Reporting summary.** Further information on research design is available in the Nature Research Reporting Summary linked to this article.

## Data availability

RNA-sequencing data supporting the findings in Figs. 2a, 4c–d, and Supplementary Fig. 15a have been deposited in GEO database and can be accessed by the public at GSE131298 (http://www.ncbi.nlm.nih.gov/geo/query/acc.cgi?token=otqvcueqttixngb&acc=GSE131298). Source data are provided with this paper.

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

## Acknowledgements

This study was supported, in part, by American Diabetes Association (ADA) 1-18-IBS-100 (to S.D.); CCHMC Pediatric Diabetes and Obesity Center (to S.D., M.H., and T.H.I.); NIH R01DK099222 (to S.D.); NIH R01DK099222-S1 (associated with S.D. and M.E.M.F. and J.R.O.); NIH R21AI139829 and R21AI139829-S1 (associated with S.D.); DoD W81XWH2010392 (associated with S.D.); Cincinnati Children's Research Foundation (CCRF) Endowed Scholar Award (to S.D.); American Heart Association (AHA) 17POST33650045 (to M.E.M.F.); ADA 1-19-PMF-019 (to M.E.M.F.); AHA 18CDA34080527 (to J.S.-G); AHA 19POST34380545 (to R.M.); NIH R01AI075159 (to C.L.K.); NIH T32AI118697 and T32GM063483-14 (associated with C.C.C.); Albert J. Ryan Foundation Fellowship (to I.T.W.H.); NIH HD07463 and GM063483 (associated with I.T.W.H.); NIH K08DK122130 (associated with S.P.W.); PHS Grant P30 DK078392 Pathology of the Digestive Disease Research Core Center at CCHMC (associated with S.D.); and CCHMC Trustee Award (to J.S.-G). Thomas C. and Joan M. Merigan Endowment at Stanford University (to D.A.R). Marie Skłodowska Curie training network "ChroMe" grant H2020-MSCA-ITN-2015-675610 (to M.H.T. and P.T.P.); German Center for Diabetes Research (DZD) (to M.H.T. and P.T.P.); Initiative and Networking Fund of the Helmholtz Association (to M.H.T. and P.T.P.); Helmholtz-Israel-Cooperation in Personalized Medicine (to P.T.P.); Helmholtz Initiative for Personalized Medicine (iMed) (to M.H.T.); and Helmholtz Portfolio Program "Metabolic Dysfunction" (to M.H.T.). We thank Dr. Burden for providing Mlc1f-cre mice. The Cd180tm1a(KOMP)Wtsi targeted ES cells used for this research project were generated by the trans-NIH Knock-Out Mouse Project (KOMP) and obtained from the KOMP Repository (www.komp.org). We thank Dr. Rajewsky, Dr. Bram, and Dr. Erickson for providing BAFF/APRIL axis transgenic mouse lines used in these studies. We also thank Dr. Daniel Giles, Matthew Lawson, and the CCHMC Pediatrics Diabetes and Obesity Center team for technical assistance.

## Author contributions

C.C.C., I.T.W.H., P.T.P, A.T., T.E.S., J.L.A., M.E.M.-F, M.S.M.A.D., J.R.O., P.C.A., J.R.D., R.M. participated in data generation. C.C.C., I.T.W.H., P.T.P, A.T., T.E.S., J.L.A., M.E.M.-F., M.S.M.A.D., J.R.O., P.C.A., J.R.D., M.J.F., L.M.F, J.S.-G., R.M., R.K., M.H., T.H.I., S.P.W., S.J.P., D.A.R., R.J.S., M.H.T., C.L.K., and S.D. participated in data analysis, interpretation, provided materials and technical support and participated in review of the manuscript. C.C.C., I.T.W.H., C.L.K., and S.D. participated in the conception and design of the study and wrote the manuscript.

## Competing interests

The authors declare the following competing interests: S.D., C.L.K. and J.L.A. hold patents on BAFF and APRIL. S.D. is a consultant for Janssen Research & Development. M.H.T. is a scientific advisor to Novo Nordisk and ERX. R.J.S. is consultant to Novo Nordisk, Sanofi, Scohia, GuidePoint Consultants, Kintai Therapeutics, and Ionis. R.J.S. receives research support or equity from Novo Nordisk, Zafgen, Astra Zeneca, Redesign Health, Ionis and Pfizer. C.C.C., I.T.W., P.F.T., A.T., T.E.S, M.E.M-F., M.S.M.A.D., J.R.O., P.C.A., J.R.D., M.J.F., L.M.F., J.S-G., R.M., R.K., M.H., T.H.I., S.P.W., S.J.P., D.A.R. declare no competing interests.

## Additional information

[1]Department of Pediatrics, The University of Cincinnati College of Medicine, Cincinnati, OH, USA. [2]Division of Immunobiology, Cincinnati Children's Hospital Medical Center, Cincinnati, OH, USA. [3]Medical Scientist Training Program, The University of Cincinnati College of Medicine and Cincinnati Children's Hospital Medical Center, Cincinnati, OH, USA. [4]Immunology Graduate Program, The University of Cincinnati College of Medicine and Cincinnati Children's Hospital Medical Center, Cincinnati, OH, USA. [5]Research Unit NeuroBiology of Diabetes, Helmholtz Center Munich, Neuherberg, Germany. [6]Institute for Diabetes and Obesity, Helmholtz Center Munich, Neuherberg, Germany. [7]German Center for Diabetes Research (DZD), Neuherberg, Germany. [8]Division of Metabolic Diseases, Technische Universität München, Munich, Germany. [9]Division of Experimental Hematology, Cincinnati Children's Hospital Medical Center, Cincinnati, OH, USA. [10]Division of Endocrinology, Cincinnati Children's Hospital Medical Center, Cincinnati, OH, USA. [11]Division of Developmental Biology, Cincinnati Children's Hospital Medical Center, Cincinnati, OH, USA. [12]Division of Gastroenterology, Hepatology and Nutrition, Cincinnati Children's Hospital Medical Center, Cincinnati, OH, USA. [13]Pediatric General and Thoracic Surgery, Cincinnati Children's Hospital Medical Center, Cincinnati, OH, USA. [14]Stem Cell & Organoid Medicine, Cincinnati Children's Hospital Medical Center, Cincinnati, OH, USA. [15]Department of Surgery, Children's Hospital Colorado, Aurora, CO, USA. [16]Columbia University Medical Center, New York, NY, USA. [17]Department of Microbiology and Immunology, Stanford University School of Medicine, Stanford, CA, USA. [18]Department of Medicine, Stanford University School of Medicine, Stanford, CA, USA. [19]Veterans Affairs Palo Alto Health Care System, Palo Alto, CA, USA. [20]Department of Surgery, Internal Medicine and Nutritional Sciences, University of Michigan, Ann Arbor, MI, USA. [21]Center for Inflammation and Tolerance, Cincinnati Children's Hospital Medical Center, Cincinnati, OH, USA. [22]Present address: Division of Rheumatology, Department of Internal Medicine and Department of Immunology & Microbiology, The University of Colorado Denver, Aurora, CO, USA. [23]Present address: University of Lausanne, Service de Pneumologie, CHUV, CLED 02.206, Epalinges, Switzerland. [24]Present address: Charlotte, NC, USA. [25]Present address: The University of North Carolina at Chapel Hill, Chapel Hill, NC, USA. [26]Present address: Chapel Hill, NC, USA. [27]Present address: National Food Institute, Technical University of Denmark, Kgs. Lyngby, Denmark. [28]Present address: Global Health Discovery & Translational Sciences, Bill & Melinda Gates Foundation, Seattle, WA, USA. [29]These authors contributed equally: Calvin C. Chan, Isaac T.W. Harley. ✉email: senad.divanovic@cchmc.org

