## [Peer Review File · Nature Communications]

Reviewer comments, first round –

Reviewer #1 (Remarks to the Author):

This study is looking at the interaction between the immune system and the adipose tissue (AT). The broad question is how immune mediator influence AT biology during diet-induced obesity. Looking at mice lacking a regulator of toll-like receptor (TLR) signalling led this team to uncover a prominent role for the cytokine BAFF in regulating weight gain and lipid handling in AT. Specifically, overexpression of BAFF in mice protects from weight gain. Gene profiling of adipocytes stimulated by BAFF revealed upregulation of pathways regulating lipid metabolism. Specifically, greater lipolysis was detected in white AT (WAT) from mice overexpressing BAFF and BAFF induced lipolysis ex-vivo. BAFF appears to augment brown AT (BAT) respiration and energy expenditure. An ortholog of BAFF, called APRIL also recapitulated the effect of BAFF on AT. BAFF+ APRIL deletion led to diet-induced obesity (DIO). Finally, decreased BMI post bariatric surgery in patients correlated with higher BAFF/APRIL levels in the serum. In conclusion the authors suggest a regulatory role for BAFF in AT and a role in controlling weight gain.

The role of BAFF in AT biology is not novel and the field is plagued with contradictory data. At least 3 published studies show opposite results to these with data suggesting that BAFF-deficiency is beneficial, reduces weight gain, possibly in females and prevents glucose intolerance.

RP105 a regulator of TLR signalling protects mice from obesity.

Fig1 shows reduced weight gain being proportional to levels of circulating BAFF in various mice and BAFF synergising with loss of RP105 in further preventing DIO. This data is speculative, many other parameters may explain this such as the activity of mice. Data from metabolic cages would have been helpful. What about the microbiome, would treatment of mice with antibiotics change the results?

Fig2: uses recombinant BAFF with no control for potential endotoxin contamination such as Boiled BAFF or a BAFF blocking antibody. Fig1F uses BAFF receptor KO mouse lines but there is no data showing whether WAT expresses any of these receptors. The receptor KO mice are not used in DIO models. Why not?

Fig 3 : same comment about missing controls for recombinant BAFF

Fig 4g : data suggest that the effect is mediated via TACI and/or BCMA , yet BCMA^{-/-} and TACI^{-/-} mice were not tested with the diet.

Stimulating AT with a single agent ex-vivo is rather meaningless as in vivo AT are stimulated by multiple factors simultaneously and AT is in contact with immune cells which stimulates additional biology.

Finally, the authors do not reconcile their data with published data contradicting these results.

Reviewer #2 (Remarks to the Author):

General assessment

The manuscript by Chan et al. is a timely study reporting the identification of novel actors to the growing bulk of molecules mediating the cross-talk between immune-related pathways and adiposity. The findings contribute to the growing body of evidence of the complex role of

inflammation-related pathways in obesity.

The main problem with the manuscript is the too limited placement of the research undertaken, parameters chosen to be analyzed and corresponding discussion in the context of the current awareness of the beige/white plasticity as essential for the control of energy expenditure, specially in rodent models. Whereas the study of BAT and brown adipocytes is essentially correct (despite some specific points, see below), the study of WAT and white adipocytes is insufficiently addressed. In the context of the current study, differential characterization of visceral (e.g. eWAT) versus subcutaneous (e.g. iWAT) in the distinct models should be shown and, for example, the extent of iWAT browning specifically addressed. All studies in vitro using white adipocytes appear to have been done in cells obtained from iWAT, which is of interest but a rather specific approach that should be analysed and discussed as representative of subcutaneous adipocyte (most beige prone) behavior in contrast with the potential behavior of adipocytes from eWAT. The combination and common discussion of data obtained from iWAT-derived adipocytes with data obtained in eWAT "in vivo" (as done, for example, in page 8, Fig 2, referring to "white adipose tissue"), is not correct in light of the current awareness of distinct browning (energy expenditure-related) and features of subcutaneous and visceral WAT (as well as distinct immunology pathways-related environment).

Overall, the manuscript needs to be improved by:

- expanding the characterization of phenotype of the mouse models to differential assessment of visceral and subcutaneous white adipose tissue, and the same with the in vitro studies using primary cultures of adipocytes from visceral versus subcutaneous depots.
- expanding the read-outs for metabolism in mice and cells to improve the soundness of some of the characterized pathways (see below).

Lastly, it's a pity that the characterization of the double-KO mice, expected to bypass redundancy processes and highly relevant to the conclusions was practically limited to body weight assessment (Fig 5I).

Specific points:

- The gross phenotype of rodent models should be expanded, and in such an "adiposity-oriented" study, gross data such as the mass (weight) of iWAT, eWAT BAT as well as liver should be provided. . These data should shown in the distinct rodent models used in a systematic manner, in the current version this is shown only at some sections.
- The metabolic characterization of models showing GTT should be complemented with ITTs (the test is described in Methods, but this reviewer couldn't find data obtained on it) and insulin levels systematically (it is shown in Fig 5, in the APRIL-/- model, but not others), basically in order to rule out primary pancreas-based insulin secretion changes originating GTT changes in the mouse models.
- Overall in the manuscript, the way of showing AUC data (e.g. in GTT data and other) is somewhat confusing when statistical differences are marked at the top (Fig 3e) at the end of the curves. Showing the histograms resulting from AUC calculations would help.
- The use of PNPLA2 and LIPE mRNAs as indicative data of lipolysis in adipose tissue, and complementary to glycerol release in cells, is not fully appropriate. It is well known that the major molecular drivers and, therefore, mediators of lipolysis intensity, are phosphorylation events on HLS, ATGL, and other rather than changes in transcript levels. Some of these parameters would be needed to strengthen the conclusions on lipolysis activity in the distinct experimental models, given the importance of this pathway in the manuscript. It is also worth mentioning that, in the current literature, assessment of lipolysis rates in vitro is often performed not only in basal but also under adrenergically-stimulated (isoproterenol, norepinephrine,...) conditions.
- Page 9, line 193: "enhanced respiration" is written whereas the corresponding Fig shows "GDP-inhibitable" respiration, which is a totally different parameters and should be described as such (and interpreted as something like "UCP1-dependent respiration", or similar).
- Page 9, line 198 on. "BAT and thermogenic parameters" is not what is shown in Suppl Fig 6a; there is no BAT data there (neither thermogenesis, in strict terms) and only whole body respiratory parameters potentially attributable to plenty of phenomena, are shown.
- The human data are interesting and of value. However, further information, easily available in

this type of cohorts, would strengthen a lot the value of the data. Sex of patients is indispensable. Insulin and the corresponding HOMA-IR levels would be of value. A number of these patients with severe obesity often show diabetes. Is this the case? Is the cohort homogeneous for diabetes condition?. Moreover, does this cohort correspond to (almost)-pediatric population?: Mean age in Suppl. Table 1 is indicated to be 16.64 year-old,...Moreover, despite data on controls are shown, features of these control individuals is lacking.

Minor points

Line 191,: possibly stating "thermogenin" as a name for UCP1 is not required, despite the use of that name years ago; it is nowadays practically in disuse.

State the age of mice used for primary cultures of white adipocytes. For brown adipocytes, "neonates" are stated,...age differences of mice being source of brown versus white adipocytes should be shown as potential source of variability in the discussion of brown-versus-white adipocyte cell comparisons in in vitro experiments.

In Figure panels showing mRNA levels (e.g. Fig 3c, Fig 4h, and so on) do state explicitly that this is "mRNA levels", stating "expression" is too imprecise.

Line 254.: replace "severely obese patients" by "patients with severe obesity" to conform with current ethical standards to refer to patients with obesity.

-

Reviewer #1 (Remarks to the Author):

We would like to start by thanking the Reviewer for his/her's insightful, thorough and constructive remarks. Our point-by-point responses to reviewer's comments are below.

This study is looking at the interaction between the immune system and the adipose tissue (AT). The broad question is how immune mediator influence AT biology during diet-induced obesity. Looking at mice lacking a regulator of toll-like receptor (TLR) signalling led this team to uncover a prominent role for the cytokine BAFF in regulating weight gain and lipid handling in AT. Specifically, overexpression of BAFF in mice protects from weight gain. Gene profiling of adipocytes stimulated by BAFF revealed upregulation of pathways regulating lipid metabolism. Specifically, greater lipolysis was detected in white AT (WAT) from mice overexpressing BAFF and BAFF induced lipolysis ex-vivo. BAFF appears to augment brown AT (BAT) respiration and energy expenditure. An ortholog of BAFF, called APRIL also recapitulated the effect of BAFF on AT. BAFF+ APRIL deletion led to diet-induced obesity (DIO). Finally, decreased BMI post bariatric surgery in patients correlated with higher BAFF/APRIL levels in the serum. In conclusion the authors suggest a regulatory role for BAFF in AT and a role in controlling weight gain.

The role of BAFF in AT biology is not novel and the field is plagued with contradictory data. At least 3 published studies show opposite results to these with data suggesting that BAFF-deficiency is beneficial, reduces weight gain, possibly in females and prevents glucose intolerance.

We fully acknowledge the existing publications focused on the intersection among BAFF, obesity, and adipose tissue¹⁻⁶. Per reviewer's suggestion we have provided an expanded and detailed discussion of aforementioned published findings. We would like to point out that the primary conclusions of existing, published, literature are centered on BAFF-driven pathogenesis of obesity-associated metabolic sequelae including glucose dysmetabolism and non-alcoholic fatty liver disease (NAFLD). However, to our knowledge, our study is the first report to depict the role of the BAFF ortholog APRIL in the context of obesogenic diet induced weight gain. Further, here we provide initial hints of how BAFF and APRIL may interplay in obesity.

Importantly, a close examination of one published study, harnessing BAFF^{-/-} mice, reveals that lack of BAFF increases body weight and weight gain³. Our findings also indicate that BAFF^{-/-} mice fed HFD exhibit increased body weight compared to WT controls (**Supplementary Fig. 9a**). Thus, this existing literature, in the context of weight gain, is supportive of our data and central hypothesis that BAFF is a modulator of weight gain.

While it is posited by Kim et. al that a gender-specific difference² may exist in BAFF modulation of glucose dysmetabolism, traditionally, female mice, unless genetically (*ob/ob*; *db/db*)⁷ or hormonally⁸ manipulated, are protected from severe diet-induced obesity (DIO) and consequent pathogenesis of obesity-associated sequelae. Congruent with this, the study published by Kim et. al demonstrates an approximate HFD-driven weight gain of 1 g (total body weight of 19 g after 4 weeks on diet) in their mouse female

cohorts². As this approach is limited, 4 weeks of HFD being unlikely to yield meaningful weight gain, and 1 g of weight gain is often not associated with obesity, it is hard to draw robust conclusions regarding gender-specific mechanisms that may underlie BAFF's effects on body weight. In addition to highlighting these limitations, in the discussion section we have now included statements that such studies should be done at thermoneutrality⁹, where female mice can become severely obese.

In sum, our data presented in this manuscript is centered through the lens of BAFF/APRIL axis on weight gain. While we have validated some of these published studies, here we have expanded our understanding of the novel role for the BAFF and its ortholog APRIL in the context of DIO. Notably, we show that: **(i)** multiple transgenic mouse lines with increased circulating BAFF levels are protected from HFD-driven weight gain including previously unpublished HFD models using BAFF-Tg and APRIL^{-/-} mice (**Fig. 1, Fig. 5**); **(ii)** BAFF^{-/-} mice exhibit increased systemic APRIL levels (**Fig. 4b**); **(iii)** APRIL similarly is sufficient to augment white adipocyte lipolysis and brown adipocyte thermogenesis (**Fig. 4e-h**); **(iv)** APRIL^{-/-} mice are protected from HFD-driven weight gain and systemic increases in circulating BAFF (**Fig. 5a-b**); and **(v)** combined deletion of BAFF and APRIL unveils robust weight gain (**Fig. 5k**).

RP105 a regulator of TLR signalling protects mice from obesity.

Fig1 shows reduced weight gain being proportional to levels of circulating BAFF in various mice and BAFF synergising with loss of RP105 in further preventing DIO. This data is speculative, many other parameters may explain this such as the activity of mice. Data from metabolic cages would have been helpful. What about the microbiome, would treatment of mice with antibiotics change the results?

We fully agree with the reviewer that additional insights into the metabolic activity of RP105^{-/-}/BAFF^{-/-} mice may unveil greater insight into how lack of BAFF in RP105^{-/-} reverses protection from DIO. Our newly generated data via TSE phenomaster system indicates that RP105^{-/-} mice exhibit significantly higher energy expenditure despite similar locomotor activity compared to WT controls (**Fig. 1f, Supplementary Fig. 3d**). Further, lack of BAFF in RP105^{-/-} mice abrogates enhancement of energy expenditure without altering locomotor activity (**Supplementary Fig. 3b-d**). Together, these data suggest that BAFF modulation of energy expenditure, within the RP105 system, may in part underlie protection from HFD-driven weight gain. Notably, such findings are in line with our overarching hypothesis that the BAFF axis is a regulator of obesogenic-diet weight gain via modulation of brown adipocyte energy expenditure and white adipocyte lipolysis.

We also thank the reviewer for his/her comments about potential impact of the microbiome. We acknowledge that BAFF is a potent modulator of B cells and T_{regs} and that both B cells and T_{regs} play an important role in the establishment of the gut microbiota^{10,11}. Our examination of the microbiome in one model of increased circulating BAFF, RP105^{-/-} mice (exhibit ~20% increase in BAFF), revealed similar intestinal microbiome between WT and RP105^{-/-} mice in both a lean and HFD-fed state (**Supplementary Fig. 1f**). These findings would suggest that the microbiome may not

play an integral role in the reversal of weight gain in RP105^{-/-}/BAFF^{-/-} mice. However, these findings could be limited to the RP105^{-/-} system. Thus, we posit that if all of the mouse lines we employed with varying levels of systemic BAFF are examined (RP105^{-/-}, RP105^{-/-}/BAFF^{-/-}, BAFF^{-/-}, μ MT, BAFF-Tg, APRIL^{-/-}, BAFF^{-/-}/APRIL^{-/-}), the microbiome would likely be different, not necessarily due to different BAFF levels but due to other altered biological processes in these specific settings, including a recent study demonstrating that BAFF modulates T_{reg} function¹¹. To carefully tease apart and draw robust conclusions in these complex systems would require additional sets of tools and complex studies including, but not limited to, microbiome depletion (e.g. Gram+ and/or Gram- cocktails in BAFF-Tg mice), microbiome transfer studies (e.g. BAFF-Tg to WT or BAFF^{-/-}/APRIL^{-/-} mice) and analysis of these mice under germ free conditions. Needless to state, completion of such studies in context of HFD feeding and analysis of all the needed metabolic and energy expenditure analysis while maintaining the desired microbiome is an enormous amount of work. While detailed analysis of the microbiome across all of our mouse lines, is clearly important, adding such findings to our existing manuscript may not significantly enhance our central message. We do however fully agree that the microbiome studies represent important future directions to further enhance our understanding of the BAFF/APRIL axis in body weight control. Thus, we have expanded our discussion to better highlight the potential impact/relevance of the microbiome in this context.

Fig2: uses recombinant BAFF with no control for potential endotoxin contamination such as Boiled BAFF or a BAFF blocking antibody. Fig1F uses BAFF receptor KO mouse lines but there is no data showing whether WAT expresses any of these receptors. The receptor KO mice are not used in DIO models. Why not?

We agree with the reviewer and apologize for this oversight. All reagents utilized in our study are ordered from companies that quantify and verify endotoxin concentration. Despite such rigor, all tissue culture or *in vivo* reagents (e.g. media, FBS, growth factors, recombinant proteins, etc) used are also additionally analyzed for endotoxin contamination via Limulus amoebocyte lysate (LAL) test in our lab – an old habit that stems from my graduate studies¹² and now a standard procedure for testing all reagents used in our studies^{9,13-15}. Our newly presented data demonstrates that endotoxin levels in our rBAFF and rAPRIL preps are not detectable (below the limit of detection [1EU/ml = 0.1 ng/ml]; Pierce LAL Kit) and likely not to have an impact on the experimental outcome (**Supplementary Fig. 6a**). Consistent with this, boiling rBAFF¹⁶ dampens its impact on white adipocyte free glycerol release (**Supplementary Fig. 6b**).

We also thank the reviewer for his/her's comments into DIO models of the BAFF/APRIL receptor deficient mice. Published literature demonstrates that adipocytes express all 3 receptors (BAFF-R, TACI, BCMA)¹⁷, however likely at levels significantly lower than that observed in B cells¹⁸. Our data indicates that mice with lack of a single receptor (BAFF-R^{-/-}, TACI^{-/-}, BCMA^{-/-}), regardless of which, are all protected from HFD-driven weight gain (**Rebuttal Fig. 1a**). Similarly, we also show that deficiency of a single receptor is sufficient to induce increased systemic levels of BAFF (**Rebuttal Fig. 1b**). TACI^{-/-} and BCMA^{-/-} mice are protected from HFD-driven weight gain, glucose dysmetabolism and hepatocellular damage compared to WT controls (**Rebuttal Fig. 1c-e**). However, despite the protection

from HFD-driven weight gain, BAFF-R^{-/-} mice exhibited similar glucose dysmetabolism and hepatocellular damage compared to WT controls (**Rebuttal Fig. 1c-e**). Together, our findings demonstrate that: **(i)** BAFF-R^{-/-6} and TACI^{-/-5} mice are protected from HFD-driven weight; and **(ii)** TACI^{-/-} mice are protected from glucose dysmetabolism⁵ are in agreement with the published reports. Conversely to our data, existing evidence suggests that BAFF-R^{-/-} mice are protected from glucose dysmetabolism but exhibit enhanced NAFLD pathogenesis⁶. If the editors and the reviewers find it necessary, we would be happy to include our novel data and/or discussion of BAFF-R^{-/-}, TACI^{-/-}, BCMA^{-/-} mice into the revised manuscript.

To our knowledge our data describing the role of BCMA in HFD-driven obesity represents the **first** depiction. Per our understanding, the lack of available BAFF-R, TACI, BCMA conditional knockout mice precludes investigation of cell-specific roles for these receptors. These deficiencies and need for future investigations on this topic are fully acknowledged and are added to our discussion in the revised manuscript. Given the large data sets and complexity of our findings, with the editor's and reviewers permission, the expanded data on the phenotype of BAFF-R^{-/-}, TACI^{-/-}, BCMA^{-/-} mice in obesity will be omitted from the current manuscript and incorporated into a subsequent manuscript.

Rebuttal Figure 1. BAFF-R^{-/-}, TACI^{-/-}, BCMA^{-/-} mice are protected from HFD-driven weight gain. BAFF-R^{-/-}, TACI^{-/-}, BCMA^{-/-} and WT mice were placed on HFD for 24 weeks. (a) Mean weight gain. (b) Systemic BAFF levels in indicated lean mice. (c) GTT at Week 18 on a HFD. (d) AUC of GTT (e) ALT at time of harvest. Representative of 3 independent experiments, n = 3-6/condition. For bar and line graphs data represents mean +/- SEM. (a-b, d-e) Unpaired two-tailed Student's t-test, (c) Area under the curve. *p < 0.05, **p < 0.01, ***p < 0.001, ****p < 0.0001.

Combined with our other findings, in this report we have identified 7 different mouse lines (RP105^{-/-}, μ MT, BAFF-Tg, APRIL^{-/-}, BAFF-R^{-/-}, TACI^{-/-}, BCMA^{-/-}) that are protected from HFD-driven weight gain, with a unifying thread of increased circulating BAFF levels.

Fig 3 : same comment about missing controls for recombinant BAFF

As above, we apologize for the oversight of the lack of appropriate controls and exclusion of potential endotoxin contamination in our BAFF stimulation studies and have provided our response to the reviewer's comment above.

Fig 4g : data suggest that the effect is mediated via TACI and/or BCMA , yet BCMA^{-/-} and TACI^{-/-} mice were not tested with the diet.

As highlighted in our response above, our data shows that mice with lack of a single receptor (BAFF-R^{-/-}, TACI^{-/-}, BCMA^{-/-}), regardless of which, are all protected from HFD-driven weight gain (**Rebuttal Fig. 1a**). These mice also exhibit increased systemic levels of BAFF (**Rebuttal Fig. 1b**). We would be happy to include our data of BAFF-R^{-/-}, TACI^{-/-}, BCMA^{-/-} mice into the manuscript if the reviewer and/or editor see it necessary.

Stimulating AT with a single agent ex-vivo is rather meaningless as in vivo AT are stimulated by multiple factors simultaneously and AT is in contact with immune cells which stimulates additional biology.

We appreciate the reviewer's comment and fully agree that the physiological milieu of adipose tissue (AT) is highly dynamic and involves multiple mediators (e.g. insulin, norepinephrine) and diverse cell types (e.g. immune cells) capable of impacting and modifying adipocyte biology. Thus, we would like to clarify that our intent was not to define BAFF as the "dominant" modulator of adipocyte biology. Rather, our novel observations simply highlight that BAFF is "an inflammatory mediator" with the capacity to skew adipocyte biology. To reductively focus our studies on understanding the "novel" abilities of BAFF axis, we chose to employ the stimulation with a single reagent in a similar fashion to other reductionist approaches.^{13,19,20} However, to show our utmost intent to improve our manuscript, in light of reviewer 1 and reviewer 2 suggestions (see below), we have generated novel data that suggest that: **(i)** Norepinephrine (NE) induces a 2 fold increase in white adipocyte free glycerol release as compared to BAFF (**Fig. 2c**) and **(ii)** BAFF does not synergize with NE to further augment white adipocyte free glycerol release (**Fig. 2c**). In addition, we have acknowledged the limitations of our reductive findings in a context of highly dynamic and intricate settings impacting adipocyte biology by improving the discussion section of the revised manuscript.

We also agree that undoubtedly BAFF impacts multiple immune cell effector capacity (e.g. Macrophages, B cells, T cells). This interplay between immune cells and AT would likely contribute to overall AT biology and function. However, the complexity of studies required to carefully dissect these mechanisms apart, and to our knowledge the lack of a well-established co-culturing system between primary adipocytes and immune cells,

would represent significant hurdles. Completion of these studies, within the context of HFD feeding, and careful analysis needed of the distinct contribution of immune cells and adipocytes is a significant amount of work in addition to our current findings. Addition of such findings to our existing manuscript may not augment our central message, which represents a first depiction of a highly intricate and complex interplay between the BAFF/APRIL axis and adipocyte biology. We fully agree that interrogation of immune cells and their impact on adipocyte biology represent key future studies that will expand the gaps in our knowledge of the BAFF/APRIL axis regulation of weight gain. We have expanded our discussion to include the potential relevance of immune cells within the AT.

Finally, the authors do not reconcile their data with published data contradicting these results.

We appreciate reviewer's comments. We apologize for this oversight and did not mean to be ambiguous. Discussion of how published studies blend with our presented findings are now highlighted in the discussion section of our manuscript. These discussion points include:

- The primary conclusions of existing studies are centered around the impact of BAFF on the pathogenesis of downstream metabolic sequelae of obesity including glucose dysmetabolism and non-alcoholic fatty liver disease (NAFLD). Our study highlights an underappreciated role for BAFF/APRIL axis in HFD-driven weight gain.
- Published studies harnessing BAFF^{-/-} mice reveals that lack of BAFF increases body weight in both male and female mice², findings in direct support of our data (**Supplementary Fig. 9a**).
- Although gender-specific difference may exist in BAFF modulation of glucose dysmetabolism, female mice used in the published study demonstrate only a 1 g weight gain on HFD, which is often not-associated with obesity. This limitation makes it hard to draw robust conclusions on gender-specific mechanisms that may underlie BAFF's effects on body weight. Future studies exploiting thermoneutral housing to unlock robust obesity in female mice to interrogate potential gender specific differences of the BAFF/APRIL axis would be fully warranted. These points will be added to the discussion within the manuscript.
- Here we have expanded our understanding of the novel role for the BAFF ortholog APRIL in the context of DIO.

Reviewer #2 (Remarks to the Author):

We would like to begin by expressing our gratitude for the reviewer's insightful remarks and constructive feedback. Our point-by-point responses to his/her comments are below.

General assessment

The manuscript by Chan et al. is a timely study reporting the identification of novel actors to the growing bulk of molecules mediating the cross-talk between immune-related pathways and adiposity. The findings contribute to the growing body of evidence of the complex role of inflammation-related pathways in obesity.

The main problem with the manuscript is the too limited placement of the research undertaken, parameters chosen to be analyzed and corresponding discussion in the context of the current awareness of the beige/white plasticity as essential for the control of energy expenditure, specially in rodent models. Whereas the study of BAT and brown adipocytes is essentially correct (despite some specific points, see below), the study of WAT and white adipocytes is insufficiently addressed. In the context of the current study, differential characterization of visceral (e.g. eWAT) versus subcutaneous (e.g. iWAT) in the distinct models should be shown and, for example, the extent of iWAT browning specifically addressed. All studies in vitro using white adipocytes appear to have been done in cells obtained from iWAT, which is of interest but a rather specific approach that should be analysed and discussed as representative of subcutaneous adipocyte (most beige prone) behavior in contrast with the potential behavior of adipocytes from eWAT. The combination and common discussion of data obtained from iWAT-derived adipocytes with data obtained in eWAT "in vivo" (as done, for example, in page 8, Fig 2, referring to "white adipose tissue"), is not correct in light of the current awareness of distinct browning (energy expenditure-related) and features of subcutaneous and visceral WAT (as well as distinct immunology pathways-related environment).

Overall, the manuscript needs to be improved by: - expanding the characterization of phenotype of the mouse models to differential assessment of visceral and subcutaneous white adipose tissue, and the same with the in vitro studies using primary cultures of adipocytes from visceral versus subcutaneous depots. - expanding the read-outs for metabolism in mice and cells to improve the soundness of some of the characterized pathways (see below).

We fully agree that differential assessment of visceral and subcutaneous WAT across our mouse models would provide additional insights. We apologize for this lack of oversight and have now included data of the distribution of iWAT, eWAT, pWAT (as applicable), BAT and Liver across all of our mouse models into the manuscript.

We also appreciate the reviewer's comments regarding characterization of adipocytes from visceral versus subcutaneous depots. As it is well-established that functional diversity exists between visceral and subcutaneous AT, and that visceral AT is a predominant contributor to metabolic derangements, we agree that elucidation of how

BAFF impacts both compartments of WAT would provide additional insights into this axis. Due to technical challenges in culturing primary adipocytes from visceral fat pads, in our lab, the labs of our collaborators, and within the field, our “*in vitro*” studies were performed on adipocytes generated from the subcutaneous AT. In our hands, SVF derived from visceral (epididymal) WAT yield reduced differentiation into adipocytes (~15-20% differentiation; **DNS**) as compared to SVF derived from subcutaneous (inguinal) WAT (~100% differentiation). Congruently, these visceral AT derived “primary adipocytes” exhibit significantly reduced norepinephrine-induced free glycerol release compared to subcutaneous WAT-derived adipocytes (**Rebuttal Fig. 2**), data consistent with poor differentiation into adipocytes.

Rebuttal Figure 2. Visceral WAT-derived adipocytes minimally release free glycerol. Primary adipocytes from WT mice derived from subcutaneous or visceral WAT and treated with saline (NS) or norepinephrine (NE; 1 μ M). Quantified free glycerol by colorimetric assay in supernatant of (a) subcutaneous-derived adipocytes and (b) visceral-derived adipocytes. A single experiment, n = 2/condition. For bar graphs data represents mean +/- SEM.

Lastly, it's a pity that the characterization of the double-KO mice, expected to bypass redundancy processes and highly relevant to the conclusions was practically limited to body weight assessment (Fig 5I).

We agree with the reviewer and apologize for the omission of the data in question. We have now generated the requested data (**Rebuttal Fig. 3**). As previously described, lack of both BAFF and APRIL unveils robust HFD-weight gain (**Fig. 5k, Rebuttal Fig. 3a**). As anticipated BAFF^{-/-}/APRIL^{-/-}

mice fed HFD exhibit: (i) increased amounts of subcutaneous (iWAT) and visceral (eWAT, pWAT) AT (**Rebuttal Fig. 3b**); (ii) decreased spleen size (**Rebuttal Fig. 3b**); (iii) protection from glucose dysmetabolism (**Rebuttal Fig. 3c-e**); and (iv) similar degree of hepatocellular damage as determined by ALT (**Rebuttal Fig. 3f**). These findings suggest that while BAFF/APRIL axis modifies HFD-driven weight gain, these mediators may modify the proinflammatory milieu in established obesity and promote glucose dysmetabolism in this setting. Notably, mice with single deletion of BAFF^{-/-} mice and APRIL^{-/-} mice similarly exhibit protection from obesity-driven glucose dysmetabolism. Hence, both BAFF and APRIL likely work in concert to impact weight gain and pathogenesis of obesity-associated sequelae. Future, detailed mechanistic investigation into the segregation of effects of the BAFF/APRIL axis on weight gain versus glucose metabolism would be fully warranted. Given the large data sets and complexity of our findings, with the editor's and reviewers permission, the expanded data on the phenotype of BAFF^{-/-}/APRIL^{-/-} mice in obesity will be incorporated into a subsequent manuscript. However, given our utmost desire to publish this current manuscript, if the editors and the reviewers find it necessary, we would be happy to include our additional data of BAFF^{-/-}/APRIL^{-/-} mice into the revised manuscript.

Rebuttal Figure 3. BAFF/APRIL axis uncouples HFD-driven weight gain from glucose dysmetabolism. BAFF^{-/-}/APRIL^{-/-} or WT mice were placed on HFD for 20 weeks. (a) Weight gain. (b) Tissue weights at time of harvest. (c) Fasting glucose at time of harvest. (d) GTT at 14 weeks on HFD. (e) AUC of GTT. (f) Systemic ALT at time of harvest. A single experiment n = 3-4/condition. (a-f) For bar graphs and line graphs data represents a mean +/- SEM. (a, d) Area under the curve. **p* < 0.05, **** *p* < 0.0001. (b-c, e-f) Unpaired two-tailed Student's t-test. **p* < 0.05.

Specific points:

- The gross phenotype of rodent models should be expanded, and in such an "adiposity-oriented" study, gross data such as the mass (weight) of iWAT, eWAT BAT as well as

liver should be provided. These data should shown in the distinct rodent models used in a systematic manner, in the current version this is shown only at some sections.

As per our response above, we fully agree that distribution of tissues would provide additional insights. We have revised our manuscript to now include data on the distribution of iWAT, eWAT, pWAT (as applicable), BAT and liver weight across all of our mouse models (**Supplementary Fig. 1a; Supplementary Fig. 3a; Supplementary Fig. 4; Supplementary Fig. 9b; Fig. 5c**).

- The metabolic characterization of models showing GTT should be complemented with ITTs (the test is described in Methods, but this reviewer couldn't find data obtained on it) and insulin levels systematically (it is shown in Fig 5, in the APRIL^{-/-} model, but not others), basically in order to rule out primary pancreas-based insulin secretion changes originating GTT changes in the mouse models.

We fully agree with the reviewer's comments and apologize for this oversight. Our overarching hypothesis is that the BAFF/APRIL axis modifies obesogenic-diet driven weight gain. As total body weight is directly tied to glucose dysmetabolism, including insulin sensitivity and insulin levels, we anticipate that our multiple mouse models protected from DIO and associated with increased BAFF would exhibit lower systemic insulin levels or insulin tolerance. Our newly generated data indicates that **(a)** RP105^{-/-} (**Supplementary Fig. 1b**) and APRIL^{-/-} (**Fig. 5i**) mice exhibit significantly lower systemic fasting insulin compared to WT controls; **(b)** HFD-fed RP105^{-/-} and μ MT mice have dampened insulin tolerance in an ITT compared to WT mice (**Rebuttal Fig. 4**). We completely concur that future, in-depth, interrogation of how the BAFF/APRIL axis impacts glucose metabolism including pancreas-mediated role(s) in insulin signaling, glucose transport, lipidomics/metabolomics would be of significant interest. We have included an expanded discussion of these topics into our manuscript.

Rebuttal Figure 4. RP105^{-/-} and μMT mice are protected from diet-induced glucose dysmetabolism. (a-b) RP105^{-/-} and WT mice were placed on HFD. (a) Insulin tolerance test (ITT) at 16 weeks. (b) Area under the curve. (c-d) μMT and WT mice were placed on HFD. (c) Insulin tolerance test (ITT) at 14 weeks. (d) Area under the curve. (a-b) Representative of 3 independent experiments, n = 6-10/condition. (c-d). Representative of 3 independent experiments, n = 5-6/condition. For bar graphs and line graphs data represents a mean +/- SEM. (a,c) Area under the curve analysis. (b,d) Unpaired two-tailed Student's t-test. **p < 0.01, ***p < 0.001.

- Overall in the manuscript, the way of showing AUC data (e.g. in GTT data and other) is somewhat confusing when statistical differences are marked at the top (Fig 3e) at the end of the curves. Showing the histograms resulting from AUC calculations would help.

We agree with the reviewer's comment and apologize for the ambiguity. We have included histograms resulting from AUC calculations for relevant observations (e.g. GTT) into the revised manuscript.

- The use of PNPLA2 and LIPE mRNAs as indicative data of lipolysis in adipose tissue, and complementary to glycerol release in cells, is not fully appropriate. It is well known that the major molecular drivers and, therefore, mediators of lipolysis intensity, are

phosphorylation events on HSL, ATGL, and other rather than changes in transcript levels. Some of these parameters would be needed to strengthen the conclusions on lipolysis activity in the distinct experimental models, given the importance of this pathway in the manuscript. It is also worth mentioning that, in the current literature, assessment of lipolysis rates in vitro is often performed not only in basal but also under adrenergically-stimulated (isoproterenol, norepinephrine,...) conditions.

We thank the reviewer for this suggestion. We acknowledge that addition of phosphorylation events in support of our existing data (e.g. HSL [aka LIPE]) would further support our conclusions. Our newly generated data indicates that treatment of adipocytes with rBAFF (**Fig. 2f**) or rAPRIL (**Fig. 4g**) also increases p-HSL protein levels, which may represent one pathway used to modify lipolysis in our system.

We also appreciate the reviewer's suggestions on assessment of lipolysis rates and fully agree that the physiological milieu of AT is highly dynamic including multiple mediators (e.g. insulin, norepinephrine) capable of impacting and modifying adipocyte biology. Per response to reviewer 1 above, we would like to clarify that our intent was not to define BAFF as the "dominant" modulator of adipocyte biology. Rather, our novel observations simply highlight that BAFF is "an inflammatory mediator" with the capacity to skew adipocyte biology. Thus, to reductively focus our studies on understanding the "novel" abilities of BAFF axis, we have chosen to employ the stimulation with a single reagent. However, to demonstrate our utmost intent to improve our manuscript, we have generated novel data that suggest that: **(i)** Norepinephrine (NE) induces a 2 fold increase in white adipocyte free glycerol release as compared to BAFF; and **(ii)** BAFF does not synergize with NE to further augment white adipocyte free glycerol release. We have also acknowledged the limitations of our reductive findings in a context of highly dynamic and intricate settings impacting adipocyte biology and expanded discussion section of our revised manuscript.

- Page 9, line 193: "enhanced respiration" is written whereas the corresponding Fig shows "GDP-inhibitable" respiration, which is a totally different parameters and should be described as such (and interpreted as something like "UCP1-dependent respiration", or similar).

We concur with the reviewer's comment and thank him/her for their suggestion. We have modified the text within the manuscript to clarify this result.

- Page 9, line 198 on. "BAT and thermogenic parameters" is not what is shown in Suppl Fig 6a; there is no BAT data there (neither thermogenesis, in strict terms) and only whole body respiratory parameters potentially attributable to plenty of phenomena, are shown.

We apologize for the lack in clarity of our message. We have corrected this in the text as per the reviewer's suggestion.

- The human data are interesting and of value. However, further information ,easily available in this type of cohorts, would strengthen a lot the value of the data. Sex of

patients is indispensable. Insulin and the corresponding HOMA-IR levels would be of value. A number of these patients with severe obesity often show diabetes. Is this the case? Is the cohort homogeneous for diabetes condition? Moreover, does this cohort correspond to (almost)-pediatric population?: Mean age in Suppl. Table 1 is indicated to be 16.64 year-old,...Moreover, despite data on controls are shown, features of these control individuals is lacking.

We greatly appreciate the reviewer's thoughtful comment and insight. We agree that inclusion of additional description of our human cohorts would help better describe our findings. We would also like to highlight that our observations stem from a pediatric cohort which provides a keen opportunity to examine the development/pathogenesis of weight from a young age. While our cohort primarily consists of obese people with significant insulin resistance (Mean fasting insulin 36.57 IU/ml, Mean HOMA-IR 9.7), most of our cohort does not exhibit Type 2 diabetes (Mean fasting glucose 85.38 mg/dL, Mean HbA1c 5.74%) (**Rebuttal Table 1**). Whether BAFF and/or APRIL may play a role in the uncoupling of weight gain from diabetes in severely obese people would yield highly valuable insight. Although of significant interest such studies in human cohorts are well beyond the purview of current manuscript. However, as there is significant interest in this area, and possible clinical relevance, we have included additional discussion of these points into the revised manuscript.

We also apologize for our oversight and have now included cohort characteristics of our lean controls into our manuscript (**Supplementary Table 2**). Our lean cohort is slightly older (Mean age 19.40) compared to our obese cohort (Mean 16.64), do not exhibit obesity (Mean BMI 22.69) nor glucose dysmetabolism (Mean HbA1c 4.85). Given our limited number of lean controls, future expanded and in-depth comparison of lean vs. persons with severe obesity may yield additional insights into the impact of the BAFF/APRIL axis in obesity development.

	Time of surgery
n	30 (4 male, 26 female)
Age (years mean +/- SEM)	16.64 +/- 0.43
BMI (mean +/- SEM)	50.15 +/- 2.25
Glucose (mean +/- SEM)	85.38 +/- 2.41
HbA1c (mean +/- SEM)	5.74 +/- 0.179
Insulin (mean +/- SEM)	36.57 +/- 4.27
HOMA-IR	9.7 +/- 1.26
Aspartate aminotransferase (mg/dL mean +/- SEM)	30.16 +/- 2.78
Alanine aminotransferase (mg/dL mean +/- SEM)	35.51 +/- 5.32
Total Cholesterol (mg/dL mean +/- SEM)	161.5 +/- 5.33
LDL (mg/dL mean +/- SEM)	92.32 +/- 4.49
HDL (mg/dL mean +/- SEM)	31.29 +/- 1.09
Triglycerides (mg/dL mean +/- SEM)	162.8 +/- 11.5

Rebuttal Table 1. Cohort Characteristics.

Minor points

Line 191,: possibly stating "thermogenin" as a name for UCP1 is not required, despite the use of that name years ago; it is nowadays practically in disuse.

We thank the reviewer for their feedback and have edited the manuscript as suggested.

State the age of mice used for primary cultures of white adipocytes. For brown adipocytes, "neonates" are stated,...age differences of mice being source of brown versus white adipocytes should be shown as potential source of variability in the discussion of brown-versus-white adipocyte cell comparisons in in vitro experiments.

We thank the reviewer for their suggestion and apologize for the ambiguity and omission of the age of mice used for primary cultures of white adipocytes. Mice aged 6-10 weeks old were utilized for isolation of primary white adipocytes. These details are now added into the materials and methods section.

Although aging plays a well-appreciated role in AT homeostasis and biology, our current study is not focused on the interplay between the BAFF/APRIL axis, weight gain, and aging. However, in light of the reviewer's constructive feedback, the potential role of aging and its impact on our proposed model has been added into the discussion of the manuscript.

In Figure panels showing mRNA levels (e.g. Fig 3c, Fig 4h, and so on) do state explicitly that this is "mRNA levels", stating "expression" is too imprecise.

We apologize for the ambiguity. The manuscript has been edited as suggested.

Line 254.: replace "severely obese patients" by "patients with severe obesity" to conform not current ethic standards to refer to patients with obesity.

We thank the reviewer for this instructive comment. The text has been modified as per reviewer's suggestion.

REFERENCES

1. Nakamura, Y., *et al.* Depletion of B cell-activating factor attenuates hepatic fat accumulation in a murine model of nonalcoholic fatty liver disease. *Sci Rep* **9**, 977 (2019).
2. Kim, B. & Hyun, C.K. Gender-Specific Mechanisms Underlying the Amelioration of High-Fat Diet-Induced Glucose Intolerance in B-Cell-Activating Factor Deficient Mice. *PLoS One* **11**, e0166225 (2016).
3. Kim, D.H. & Do, M.S. BAFF knockout improves systemic inflammation via regulating adipose tissue distribution in high-fat diet-induced obesity. *Exp Mol Med* **47**, e129 (2015).
4. Kim, Y.H., Choi, B.H., Cheon, H.G. & Do, M.S. B cell activation factor (BAFF) is a novel adipokine that links obesity and inflammation. *Exp Mol Med* **41**, 208-216 (2009).
5. Liu, L., *et al.* TACI-Deficient Macrophages Protect Mice Against Metaflammation and Obesity-Induced Dysregulation of Glucose Homeostasis. *Diabetes* **67**, 1589-1603 (2018).
6. Kawasaki, K., *et al.* Blockade of B-cell-activating factor signaling enhances hepatic steatosis induced by a high-fat diet and improves insulin sensitivity. *Lab Invest* **93**, 311-321 (2013).
7. Anstee, Q.M. & Goldin, R.D. Mouse models in non-alcoholic fatty liver disease and steatohepatitis research. *Int J Exp Pathol* **87**, 1-16 (2006).
8. Lutz, T.A. & Woods, S.C. Overview of animal models of obesity. *Curr Protoc Pharmacol* **Chapter 5**, Unit5 61 (2012).
9. Giles, D.A., *et al.* Thermoneutral housing exacerbates nonalcoholic fatty liver disease in mice and allows for sex-independent disease modeling. *Nat Med* **23**, 829-838 (2017).
10. Shulzhenko, N., *et al.* Crosstalk between B lymphocytes, microbiota and the intestinal epithelium governs immunity versus metabolism in the gut. *Nat Med* **17**, 1585-1593 (2011).
11. Stohl, W. & Yu, N. Promotion of T Regulatory Cells in Mice by B Cells and BAFF. *J Immunol* (2020).
12. Divanovic, S., *et al.* Negative regulation of Toll-like receptor 4 signaling by the Toll-like receptor homolog RP105. *Nat Immunol* **6**, 571-578 (2005).
13. Cappelletti, M., *et al.* Type I interferons regulate susceptibility to inflammation-induced preterm birth. *JCI Insight* **2**, e91288 (2017).
14. Allen, J.L., *et al.* Cutting edge: regulation of TLR4-driven B cell proliferation by RP105 is not B cell autonomous. *J Immunol* **188**, 2065-2069 (2012).
15. Harley, I.T., *et al.* IL-17 signaling accelerates the progression of nonalcoholic fatty liver disease in mice. *Hepatology* **59**, 1830-1839 (2014).
16. Batten, M., *et al.* BAFF mediates survival of peripheral immature B lymphocytes. *J Exp Med* **192**, 1453-1466 (2000).
17. Alexaki, V.I., *et al.* Adipocytes as immune cells: differential expression of TWEAK, BAFF, and APRIL and their receptors (Fn14, BAFF-R, TACI, and

- BCMA) at different stages of normal and pathological adipose tissue development. *J Immunol* **183**, 5948-5956 (2009).
18. Novak, A.J., *et al.* Expression of BCMA, TACI, and BAFF-R in multiple myeloma: a mechanism for growth and survival. *Blood* **103**, 689-694 (2004).
 19. Fujita, Y., *et al.* Deficient leptin signaling ameliorates systemic lupus erythematosus lesions in MRL/Mp-Fas lpr mice. *J Immunol* **192**, 979-984 (2014).
 20. Smith, M.A., *et al.* Calcium Channel CaV2.3 Subunits Regulate Hepatic Glucose Production by Modulating Leptin-Induced Excitation of Arcuate Pro-opiomelanocortin Neurons. *Cell Rep* **25**, 278-287 e274 (2018).

Reviewer comments, second round –

Reviewer #2 (Remarks to the Author):

This is the report to the revised version of the manuscript "A BAFF/APRIL axis regulates obesogenic-diet driven weight gain" by Chan et al.

The main points raised in the previous reviewing of the manuscript were:

- The necessity to characterize not only indications of thermogenic activation of BAT but also the extent of browning of WAT "in vivo" in the mouse models of intervention used. Despite the general comments in the first paragraph of Rebuttal message, this reviewer is unable to find any significant further contribution after this request in any of the multiple animal models used (e.g. microscopy to assess extent of browning of iWAT, gene expression of thermogenic/beige markers in iWAT, UCP1 protein levels,....). that allow to check this key process in this experimental context. The data in WAT remain restricted to some parameters related to lipolysis. In the context of the current state-of-the-art of knowledge of the relationship between adipose tissue plasticity and energy expenditure in rodent models and the data obtained of altered energy expenditure, the lack of data on WAT browning throughout the manuscript remains a substantial weakness. Moreover, that makes that some general statements should be amended (e.g. in heading at line 203, " in BAT" should be added).
- The necessity to enhance the distinctions between findings in iWAT and eWAT given the distinct and often opposite biology of these depots in relation to the inflammation/adiposity relationships. The authors explain the lack of differentiation capacity of precursors from eWAT relative to iWAT, which appears reasonable for focusing their research in vitro to the behavior of subcutaneous white adipocytes. However, across the manuscript this specificity is largely ignored and many sentences in Results and Discussion quote "in white adipocytes" in a manner that may lead to the reader to a not totally correct perception. The manuscript should be revised to add at key points "subcutaneous" to provide a correct description of the actual findings.
- The necessity to expand the characterization of the effects of the double BAFF/APRIL KO. The authors introduced as relevant the model of double BAFF/APRIL as part of their report in the original manuscript and the mere provision of weight data for the model was found insufficient by this reviewer. The data in Rebuttal Fig and the accompanying reasoning accomplish to a reasonable extent the requested missing data and should be added to the manuscript. This reviewer does not understand the request not to introduce the data in the rebuttals Fig 3 provided, as this is key not to keep the description of a rodent model in the manuscript practically with only body weight data provided.
- Necessity to clarify and complement human data. In this section the main problems remaining are the limited quality of the cohorts studies and the overstatements from the data obtained that are quoted, for example, in the heading at line 276. The cohort of patients studied is practically pediatric and highly sex-biased (4 male, 26 female, according to the Table as provided in the Rebuttal letter, this sex distribution data are missing in the Supplementary Table in the manuscript and should be added); the control cohort of healthy individuals used for comparison is very limited in number (10 according to Supplementary Table, but only 4 (!) according to the Fig 6; mean age is not dramatically different but it is highly significantly different between both cohorts , sex distribution is not provided in the control cohort,...The rationale for exploring the role of BAFF and APRIL using human adipocytes from severely obese patients and not regular human adipocytes is unclear. This reviewer understands that the human data at the end of the manuscript are complementary to the bulk of the studies reported previously in mouse models. However, a standard of quality of cohorts for comparisons is required before reaching conclusions in humans, and several of the characteristics of the cohorts mentioned above do not appear to provide these standards of quality. If these data are to be added to the manuscript, the description of results should be rewritten stressing that these are very preliminary data, make explicit the limitations and avoid general overstatements as , for example, in heading 276.

Reviewer #3 (Remarks to the Author):

Comments on the Rebuttal to Reviewer 1

If one simply distills this complex paper into the tenant that increased BAFF protects from HFD weight gain and loss of BAFF has the opposite effect, the authors are correct in saying that multiple studies have shown that loss of BAFF leads to increased weight gain and hence consistent with the thrust of this work. The newly included data in supp figure 9 showing increased weight gain in the BAFF^{-/-} mouse is a bit mysterious as the starting weights of WT and BAFF^{-/-} are different yet the slopes of the weight gain are roughly parallel and how are these data different from those in fig 4a- i.e. what is the actual conclusion? I find all the published studies with BAFF^{-/-} mice show a rather marginal effect. Now the protection seen with deletion of the BAFF/ARPIL receptor TACI or BCMA is seriously inconsistent with the above premise yet consistent with a prior study (ref 40 in the manuscript). Authors seem to imply that loss of the receptor increases BAFF levels and hence the protection? Yet these elevations are tiny (~2-3x) compared to the elevation in uMT and BAFF-Tg (10-100x) and begs the question is this small change really capable of reducing the weight gain. I find this topic very complicated and a central picture does not clearly emerge from the paper.

Novelty was challenged given the multiple publications on this topic and the authors point to the inclusion of the APRIL side of problem which was unexplored in the prior BAFF^{-/-} studies as the novel element. While this point is true it is not especially surprising given the large level of redundancy of the two ligands and to actually dissect the problem properly, one needs to explore the receptor side of BAFF/APRIL axis. While they have some data, they stated that they want to retain these data for another paper given the size of the data set. I think some data are needed to resolve the conflicting aspects of the ligand vs receptor KO data.

Regarding the issue of mucosal microbiome changes in the various genetically modified mouse lines, the authors present data for their keystone observation on the RP105^{-/-} mice showing little gross modulation of the microbiome. While a careful dissection of this issue is outside the scope of this paper, it is in my view a weakness in their conclusion that BAFF levels in vivo are directly affecting the status of the adipose tissue as opposed to indirect effects. The direct effects of BAFF on cultured adipose tissue may not be underlying the in vivo situation, e.g. changes in RNA levels shown in fig 2e are very small. Again, while outside the scope of this paper, there could be major changes in the gut status perhaps not reflected in microbial distributions. For example, BAFF-Tg mice have massive increases in IgA and the loss of containment of commensal Ig reactivity and plasma cells within the mucosal compartment (McCarthy JCI 2011). The excessive BAFF levels in BAFF-Tg leads to highly oligomerized BAFF that can mimic APRIL and therefore one would expect TACI and/or BCMA to be the critical receptor (as they observe), if indeed there is a potential gut component to this axis. A provocative and perhaps related finding is that BAFF-Tg mice were totally protected from EAE and a suppressive effect of plasma cells was proposed as the mechanism (Rojas 2019 Cell).

Other than the issue of novelty for a high level journal and the conflicting receptor KO issues, the authors have rebutted the remaining points effectively.

Sources of the KO and Tg mice should be noted in the methods.

Reviewer #2 (Remarks to the Author):

We would like to start by thanking the Reviewer for his/her thoughtful, insightful, and constructive feedback. Our point-by-point responses to reviewer's comments are below.

- The necessity to characterize not only indications of thermogenic activation of BAT but also the extent of browning of WAT "in vivo" in the mouse models of intervention used.

Despite the general comments in the first paragraph of Rebuttal message, this reviewer is unable to find any significant further contribution after this request in any of the multiple animal models used (e.g. microscopy to assess extent of browning of iWAT, gene expression of thermogenic/beige markers in iWAT, UCP1 protein levels,...). that allow to check this key process in this experimental context. The data in WAT remain restricted to some parameters related to lipolysis. In the context of the current state-of-the-art of knowledge of the relationship between adipose tissue plasticity and energy expenditure in rodent models and the data obtained of altered energy expenditure, the lack of data on WAT browning throughout the manuscript remains a substantial weakness. Moreover, that makes that some general statements should be amended (e.g. in heading at line 203, " in BAT" should be added).

We are appreciative of the reviewer's feedback. Our additional novel findings demonstrate that iWAT mRNA expression of *Ucp1* dependent thermogenic/beige markers (e.g. *Pgc1a*, *Ucp1*, *Dio2*, *Cidea*, *Elovl6*) are largely similar between WT and examined transgenic mice of the BAFF/APRIL axis (**Supplementary Fig. 9**, **Supplementary Fig. 16f**). Combined with our previously presented *in vitro* data in **Supplementary Fig. 8b**, this would suggest that that rBAFF may be insufficient to induce subcutaneous (inguinal) white adipocyte *Ucp1* expression and that the BAFF/APRIL axis may not impact *Ucp1*-dependent beiging of subcutaneous WAT. Given the complexity of the BAFF/APRIL system we fully acknowledge that future in-depth interrogation of whether BAFF/APRIL axis modifies *Ucp1*-dependent and independent subcutaneous WAT beiging *in vivo* would be of significant interest including microscopic tissue assessment and analysis of subcutaneous adipose tissue/adipocyte respiration. In addition, as the dependency of *Ucp1* deletion on weight gain is uncovered at thermoneutrality¹, future examination of BAFF/APRIL axis at thermoneutrality might be warranted. In line with our new findings and conclusions, key statements regarding WAT browning have been inserted within the manuscript. We have expanded the discussion section to include these points in an attempt to enhance reader interpretation of our studies.

- The necessity to enhance the distinctions between findings in iWAT and eWAT given the distinct and often opposite biology of these depots in relation to the inflammation/adiposity relationships.

The authors explain the lack of differentiation capacity of precursors from eWAT relative to iWAT, which appears reasonable for focusing their research in vitro to the behavior of

subcutaneous white adipocytes. However, across the manuscript this specificity is largely ignored and many sentences in Results and Discussion quote "in white adipocytes" in a manner that may lead to the reader to a not totally correct perception. The manuscript should be revised to add at key points "subcutaneous" to provide a correct description of the actual findings.

We appreciate the reviewer's comment and are happy that he/she agrees with our rationale for utilizing subcutaneous white adipocytes in our studies. We fully agree and have added "subcutaneous" at key points in the manuscript to clarify the description of our findings.

- The necessity to expand the characterization of the effects of the double BAFF/APRIL KO.

The authors introduced as relevant the model of double BAFF/APRIL as part of their report in the original manuscript and the mere provision of weight data for the model was found insufficient by this reviewer. The data in Rebuttal Fig and the accompanying reasoning accomplish to a reasonable extent the requested missing data and should be added to the manuscript. This reviewer does not understand the request not to introduce the data in the rebuttals Fig 3 provided, as this is key not to keep the description of a rodent model in the manuscript practically with only body weight data provided.

We thank the reviewer for his/her encouragement to include this data into the manuscript. As requested, these data are now presented in **Figure 6** and we have expanded our discussion to reflect inclusion of these data.

- Necessity to clarify and complement human data.

In this section the main problems remaining are the limited quality of the cohorts studies and the overstatements from the data obtained that are quoted, for example, in the heading at line 276. The cohort of patients studied is practically pediatric and highly sex-biased (4 male, 26 female, according to the Table as provided in the Rebuttal letter, this sex distribution data are missing in the Supplementary Table in the manuscript and should be added); the control cohort of healthy individuals used for comparison is very limited in number (10 according to Supplementary Table, but only 4 (!) according to the Fig 6; mean age is not dramatically different but it is highly significantly different between both cohorts , sex distribution is not provided in the control cohort,...The rationale for exploring the role of BAFF and APRIL using human adipocytes from severely obese patients and not regular human adipocytes is unclear. This reviewer understands that the human data at the end of the manuscript are complementary to the bulk of the studies reported previously in mouse models. However, a standard of quality of cohorts for comparisons is required before reaching conclusions in humans, and several of the characteristics of the cohorts mentioned above do not appear to provide these standards of quality. If these data are to be added to the manuscript, the description of results should be rewritten stressing that these are very preliminary data,

make explicit the limitations and avoid general overstatements as , for example, in heading 276.

We agree with the reviewer's feedback and are happy that the human data generated significant interest. We fully acknowledge the limited lean cohort presented in this study. The limitation of our findings are now directly highlighted in the text -- something that provides additional transparency of our findings. However, our intent is to present *highly preliminary* evidence invoking the human relevance for the BAFF/APRIL axis as a potential clinical predictor and marker for weight loss outcomes post-bariatric surgery. Recruitment of lean controls is significantly limited in our IRB-approved clinical cohort - even more so now given the COVID-19 situation. Modifications of IRB protocols in the current environment are extremely difficult, if not impossible. Of note, future and thorough exploration of the BAFF/APRIL axis in humans (e.g. expanded numbers of lean controls, exploration of pediatric vs. adult obese persons, sex-dependent factors) are warranted to generate the most robust conclusions. We have included key statements and modified our discussion to soften our conclusions and further amplify the limitations and preliminary nature of our results.

Reviewer #3 (Remarks to the Author):

We would like to express our gratitude for the reviewer's thorough remarks. Our point-by-point responses to reviewer's comments are below.

If one simply distills this complex paper into the tenant that increased BAFF protects from HFD weight gain and loss of BAFF has the opposite effect, the authors are correct in saying that multiple studies have shown that loss of BAFF leads to increased weight gain and hence consistent with the thrust of this work. The newly included data in supp figure 9 showing increased weight gain in the BAFF^{-/-} mouse is a bit mysterious as the starting weights of WT and BAFF^{-/-} are different yet the slopes of the weight gain are roughly parallel and how are these data different from those in fig 4a- i.e. what is the actual conclusion? I find all the published studies with BAFF^{-/-} mice show a rather marginal effect. Now the protection seen with deletion of the BAFF/ARL101 receptor TAC1 or BCMA is seriously inconsistent with the above premise yet consistent with a prior study (ref 40 in the manuscript). Authors seem to imply that loss of the receptor increases BAFF levels and hence the protection? Yet these elevations are tiny (~2-3x) compared to the elevation in uMT and BAFF-Tg (10-100x) and begs the question is this small change really capable of reducing the weight gain. I find this topic very complicated and a central picture does not clearly emerge from the paper.

Novelty was challenged given the multiple publications on this topic and the authors point to the inclusion of the APRIL side of problem which was unexplored in the prior BAFF^{-/-} studies as the novel element. While this point is true it is not especially surprising given the large level of redundancy of the two ligands and to actually dissect the problem properly, one needs to explore the receptor side of BAFF/APRIL axis. While they have some data, they stated that they want to retain these data for another paper given the size of the data set. I think some data are needed to resolve the conflicting aspects of the ligand vs receptor KO data.

Regarding the issue of mucosal microbiome changes in the various genetically modified mouse lines, the authors present data for their keystone observation on the RP105^{-/-} mice showing little gross modulation of the microbiome. While a careful dissection of this issue is outside the scope of this paper, it is in my view a weakness in their conclusion that BAFF levels in vivo are directly affecting the status of the adipose tissue as opposed to indirect effects. The direct effects of BAFF on cultured adipose tissue may not be underlying the in vivo situation, e.g. changes in RNA levels shown in fig 2e are very small. Again, while outside the scope of this paper, there could be major changes in the gut status perhaps not reflected in microbial distributions. For example, BAFF-Tg mice have massive increases in IgA and the loss of containment of commensal Ig reactivity and plasma cells within the mucosal compartment (McCarthy JCI 2011). The excessive BAFF levels in BAFF-Tg leads to highly oligomerized BAFF that can mimic APRIL and therefore one would expect TAC1 and/or BCMA to be the critical receptor (as they observe), if indeed there is a potential gut component to this axis. A provocative and perhaps related finding is that BAFF-Tg

mice were totally protected from EAE and a suppressive effect of plasma cells was proposed as the mechanism (Rojas 2019 Cell).

We appreciate the reviewer's feedback and comments. We are happy that our HFD studies in BAFF-R^{-/-}, TACI^{-/-}, and BCMA^{-/-} mice generated significant interest from the reviewer and per his/her encouragement have included this data into our manuscript (**Supplementary Figure 16**). While conditional BAFF-R knockout mouse have been generated by another investigator^{2,3}, to our knowledge conditional knockout TACI and BCMA mice are not currently available. It remains technically challenging, particularly in the current COVID-19 pandemic, to obtain and/or generate the tools necessary to closely interrogate the contribution of each individual receptor. We are fully committed to pursue such studies (via generation of these mouse strains) in subsequent future manuscripts.

We apologize for the ambiguity of the BAFF^{-/-} weight data. To clarify, the data presented in **Fig. 4a** depicts *no difference in weight gain* after HFD challenge between BAFF^{-/-} and WT counterparts. As the reviewer correctly points out, this is due to the roughly parallel slopes of *total body weight* as depicted in **Supplementary Fig. 11b**. However, BAFF^{-/-} mice exhibited inherently larger total body weight (**Supplementary Fig. 11b**). We believe that this may be a consequence of activation of compensatory mechanisms leading to increased APRIL levels in BAFF^{-/-} mice (**Fig. 4b**).

We completely agree that the biology presented is highly complex as it consists of two ligands (BAFF and APRIL), three receptors (BAFF-R, TACI, BCMA) and highly functional redundancy and compensatory biology (deletion of BAFF or APRIL drives augmented expression of the opposite ligand) that could be differentially displayed depending on the tissue/cell type function in these settings. As the reviewer stated, the central message of our story is that the BAFF/APRIL axis modifies HFD-driven weight gain. To underscore this central message, to date we have tested 7 different transgenic mouse lines with increased levels of BAFF that are associated with *protection* from HFD-driven weight gain (RP105^{-/-}, BAFF-Tg, μ MT, APRIL^{-/-} [**Fig. 1a; Fig 1h; Fig. 5b**]; BAFF-R^{-/-}, TACI^{-/-}, BCMA^{-/-} [**Supplementary Figure 16a**]). In contrast, we have examined 3 different transgenic mouse lines lacking BAFF that are associated with *increased* weight (RP105^{-/-}/BAFF^{-/-}, BAFF^{-/-}, BAFF^{-/-}/APRIL^{-/-} [**Fig. 1b; Supplementary Fig. 11b; Fig. 6a**]). Across our examination of these 10 mouse lines, the unifying feature is associated with varying levels of BAFF. As highlighted by the reviewer, while outside the scope of this manuscript, we completely agree that additional variable factors (e.g. microbiome, immune/inflammatory milieu, BAFF/APRIL axis receptor expression across various different tissues) across these models likely contribute to varying degrees on the modification of HFD-driven weight. In fact, our studies highlight the necessity for future in-depth interrogation to more carefully dissect apart this complexity. Importantly, the limitations of our findings have been included in the revised discussion section of the manuscript.

We thank the reviewer for his/her comments that our findings regarding APRIL are novel. We entirely concur that functional redundancy likely exists given that both BAFF and APRIL can bind to BAFF-R/TACI/BCMA depending on their monomeric or multimeric conformation. In the preliminary studies presented in our manuscript, we attempted to

remove this functional redundancy through utilization of BAFF^{-/-}/APRIL^{-/-} mice which led to significant (~50%) increase in HFD-driven weight gain (**Fig. 6a**). Additional data supporting this argument was previously presented in (**Fig. 2g**) suggesting that the lack of BAFF-R or TACI or BCMA abrogates rBAFF-driven increase in white adipocyte lipolysis. Conversely only lack of TACI or BCMA dampens rAPRIL-driven increase in white adipocyte lipolysis (**Fig. 4h**). These findings suggest that: **(a)** a functional redundancy may exist between BAFF/APRIL and the three receptors which in turn compensates for the HFD-driven weight gain and subcutaneous white adipocyte lipolysis; and **(b)** previously established selectivity of BAFF and APRIL binding to these receptors in immune cells is likely conserved in white adipocytes.

To more closely interrogate the contribution of these receptor(s) to white adipocyte lipolysis, our new studies demonstrate that lack of BAFF-R and BCMA or lack of all three receptors (BAFF-R, TACI, and BCMA) abrogates rBAFF (**Supplementary Fig. 7**) and rAPRIL (**Supplementary Fig. 13**) driven lipolysis. These new findings would indicate that TACI alone may be insufficient to mediate BAFF and/or APRIL-driven subcutaneous white adipocyte lipolysis. This raises the possibility that BAFF-R, BCMA or the engagement of multiple receptors are needed for the induction of lipolysis. Full in-depth interrogation requiring tissue/cell-specific deletion of BAFF-R, TACI and BCMA to clarify their role in BAFF/APRIL axis regulation of weight gain is completely warranted in future studies. As noted above, while publications indicate availability of a conditional BAFF-R knockout mouse, to our knowledge conditional knockout TACI and BCMA mice are not currently available.

Thus, in-depth exploration of the receptor side of the BAFF/APRIL axis are clearly warranted – in particular if specific receptor(s) can be exploited as effective preventative and/or therapeutic method for modifying weight gain.

Other than the issue of novelty for a high level journal and the conflicting receptor KO issues, the authors have rebutted the remaining points effectively.

We appreciate the reviewer's comments that we have effectively rebutted the remaining points.

Sources of the KO and Tg mice should be noted in the methods.

We thank the reviewer for their comment and apologize for the oversight. We have inserted the necessary information into the manuscript.

REFERENCES

1. Feldmann, H.M., Golozoubova, V., Cannon, B. & Nedergaard, J. UCP1 ablation induces obesity and abolishes diet-induced thermogenesis in mice exempt from thermal stress by living at thermoneutrality. *Cell Metab* **9**, 203-209 (2009).
2. Ng, L.G., *et al.* B cell-activating factor belonging to the TNF family (BAFF)-R is the principal BAFF receptor facilitating BAFF costimulation of circulating T and B cells. *J Immunol* **173**, 807-817 (2004).
3. Muller-Winkler, J., *et al.* Critical requirement for BCR, BAFF, and BAFFR in memory B cell survival. *J Exp Med* **218**(2021).

Reviewer comments, third round –

Reviewer #2 (Remarks to the Author):

In relation to the revised manuscript by Divanovic and collaborators, the authors have advanced significantly in fulfilling the points raised in the previous reviewing report. Explicit statements on subcutaneous adipose tissue, limitations of the human studies and inclusion of data in Fig 6 are appropriate modifications.

My only remaining point is in relation to providing the data on the extent of WAT browning. The way the data are shown as heat-map in Supplemental Fig 9 does not appear to be the most appropriate and may raise some doubts to readers about the negative statements of WAT browning in some mouse models, to the view of this reviewer. The legend states "Representative of 3 independent experiments, n= 4/condition. For heat maps, data represents mean. Unpaired two-tailed". Under these conditions, expression of the means and SD/SEM bars instead of heat map would be desirable, as in most of the manuscript Figs. There are plenty of experiments with N = 4 in the manuscript shown as conventional means + SEM in the paper, so it may be better to do the same. My advise would be to modify the presentation of data to improve clarity.

Reviewer #4 (Remarks to the Author):

This is a revised version of a well-written, novel and elegantly presented manuscript that dissects the impact of two structurally and functionally similar immune mediators called BAFF and APRIL on metabolism, including gain weight and adipose tissue function. I feel that authors adequately addressed the main Reviewers' concerns. I only have a minor comment.

Discussion, page 17: "Further, BAFF-Tg mice exhibit enhanced IgA levels and loss of plasma cells in the mucosal compartment of the gut". I would encourage authors to revisit this statement, as the enhanced gut IgA secretion in BAFF-Tg mice should result from an expansion (not loss) and/or increased antibody secretion of IgA+ plasma cells in the intestinal lamina propria (LP). I would also change ref. 66, which does not specifically deal with plasma cells from the intestinal LP of BAFF-Tg mice.

Reviewer #2 (Remarks to the Author):

We would like to start by thanking the Reviewer for his/her insightful and constructive feedback. Our point-by-point responses to reviewer's comments are below.

In relation to the revised manuscript by Divanovic and collaborators, the authors have advanced significantly in fulfilling the points raised in the previous reviewing report. Explicit statements on subcutaneous adipose tissue, limitations of the human studies and inclusion of data in Fig 6 are appropriate modifications.

My only remaining point is in relation to providing the data on the extent of WAT browning. The way the data are shown as heat-map in Supplemental Fig 9 does not appear to be the most appropriate and may raise some doubts to readers about the negative statements of WAT browning in some mouse models, to the view of this reviewer. The legend states "Representative of 3 independent experiments, n= 4/condition. For heat maps, data represents mean. Unpaired two-tailed". Under these conditions, expression of the means and SD/SEM bars instead of heat map would be desirable, as in most of the manuscript Figs. There are plenty of experiments with N = 4 in the manuscript shown as conventional means + SEM in the paper, so it may be better to do the same. My advise would be to modify the presentation of data to improve clarity.

We appreciate the reviewer's thoughtful comment and are happy we have advanced significantly previous points raised. We concur that data presented as heat maps in Supplementary Figure 9 and Supplementary Figure 16 may lead to some ambiguity and apologize for this oversight. We have modified the presentation of this data to a conventional bar graph displaying replicate data points to enhance reader clarity (**Supplementary Figure 9 and now Supplementary Figure 16f-j**).

Reviewer #4 (Remarks to the Author):

We would like to express our appreciation for the reviewer's thoughtful remarks. Our point-by-point responses to reviewer's comments are below.

This is a revised version of a well-written, novel and elegantly presented manuscript that dissects the impact of two structurally and functionally similar immune mediators called BAFF and APRIL on metabolism, including gain weight and adipose tissue function. I feel that authors adequately addressed the main Reviewers' concerns. I only have a minor comment.

Discussion, page 17: "Further, BAFF-Tg mice exhibit enhanced IgA levels and loss of plasma cells in the mucosal compartment of the gut". I would encourage authors to revisit this statement, as the enhanced gut IgA secretion in BAFF-Tg mice should result from an expansion (not loss) and/or increased antibody secretion of IgA+ plasma cells in the intestinal lamina propria (LP). I would also change ref. 66, which does not specifically deal with plasma cells from the intestinal LP of BAFF-Tg mice.

We thank the reviewer for his/her comments and are glad that he/she feels that main Reviewers' concerns have been adequately addressed. We also thank the reviewer for pointing out the discrepancy of our discussion point on the relationship between IgA levels in BAFF-Tg mice and apologize for the confusion and misstatement. As referenced, McCarthy et. al previously demonstrated that BAFF-Tg mice exhibit increased systemic levels of IgA. Further this augmentation of serum IgA was found to be dependent on commensal flora in BAFF-Tg mice. We entirely agree that this is likely mediated by a local, gut mucosal, expansion in either plasma cells or memory B cells. Whether and how systemic IgA, in the setting of the BAFF axis, affects intestinal homeostasis and consequently contributes to obesity and weight gain remains undefined. We have modified our discussion to more accurately convey this message.